# HillTau: A fast, compact abstraction for model reduction in biochemical signaling networks

**Upinder S. Bhalla***

NCBS-TIFR, Bangalore, Karnataka, India

* bhalla@ncbs.res.in

## Abstract

Signaling networks mediate many aspects of cellular function. The conventional, mechanistically motivated approach to modeling such networks is through mass-action chemistry, which maps directly to biological entities and facilitates experimental tests and predictions. However such models are complex, need many parameters, and are computationally costly. Here we introduce the HillTau form for signaling models. HillTau retains the direct mapping to biological observables, but it uses far fewer parameters, and is 100 to over 1000 times faster than ODE-based methods. In the HillTau formalism, the steady-state concentration of signaling molecules is approximated by the Hill equation, and the dynamics by a time-course *tau*. We demonstrate its use in implementing several biochemical motifs, including association, inhibition, feedforward and feedback inhibition, bistability, oscillations, and a synaptic switch obeying the BCM rule. The major use-cases for HillTau are system abstraction, model reduction, scaffolds for data-driven optimization, and fast approximations to complex cellular signaling.

**Data Availability Statement:** All relevant data are within the manuscript and its Supporting Information files. Source code and documentation

## Author summary

Chemical signals mediate many computations in cells, from housekeeping functions in all cells to memory and pattern selectivity in neurons. These signals form complex networks of interactions. Computer models are a powerful way to study how such networks behave, but it is hard to get all the chemical details for typical models, and it is slow to run them with standard numerical approaches to chemical kinetics. We introduce HillTau as a simplified way to model complex chemical networks. HillTau models condense multiple reaction steps into single steps defined by a small number of parameters for activation and settling time. As a result the models are simple, easy to find values for, and they run quickly. Remarkably, they fit the full chemical formulations rather well. We illustrate the utility of HillTau for modeling several signaling network functions, and for fitting complicated signaling networks.

This is a *PLOS Computational Biology* Methods paper.

are available at https://github.com/BhallaLab/HillTau.

**Funding:** USB received support from NCBS-TIFR which is supported by the Department of Atomic Energy, Government of India, under project identification No. RTI 4006. USB also received grant support from Department of Science and Technology, project No. DST/INT/SWD/VR/P-09/2016. The funders had no role in study design, data collection and analysis, decision to publish or preparation of the manuscript.

**Competing interests:** The authors have declared that no competing interests exist.

# Introduction

John von Neumann's elephant haunts mechanistically detailed models. von Neumann was reported to have claimed that he could fit an elephant with 4 parameters, with the implication that models with many parameters are under-constrained and over-fitted [1]. There are two major arguments to exorcise this elephant: that mechanistic detail is needed to address certain kinds of questions; and that in the era of big data it is both easier and less biased to simply build up detailed models with all the available pieces. Here we describe a model formalism, the HillTau form, to help navigate between biological mechanisms and big data on the one hand, and the desirability of condensed model representations that expose the key principles of system function.

Cellular, and particularly synaptic signaling, is notoriously complex. There are an estimated 1400 protein species localized to the postsynaptic density alone [2]. These support a range of functions including synaptic transmission, maintenance, plasticity, activity-driven protein synthesis, metabolic control, and traffic [3].

Mass-action chemistry is a common denominator for mechanistically inspired modeling of these phenomena. This has the key virtue of defining specific biological entities (molecules) and processes (reactions) that map directly to experimental observables. Many studies are based at this level [4,5]. Further levels of mechanistic detail include reaction-diffusion, stochastic chemistry mesoscopic stochastic methods with trapezoidal or cubic meshes [6,7] and even single-particle reaction-diffusion calculations [8,9]. Note that the additional mechanistic detail comes at a considerable computational cost.

A few studies have found ways to lessen the level of detail, typically by focusing on interactions without dynamics (e.g, [10]) or on dynamics with highly reduced interactions (e.g., [11]). Model detail may also be abstracted out through model reduction, which starts from a detailed (usually mass-action or Michaelis-Menten ODE form) model and strips it down to core interactions needed to account for model behavior. Another reduction approach is to identify 'fast' reactions in the system, which settle much faster than the overall system, and can be replaced with algebraic relations [12,13]. These are a subset of general approaches to model reduction using quasi-equilibrium and quasi-steady state methods (reviewed in [14]). There are serveral other model reduction techniques (reviewed by Snowden [15]). Most of these methods retain the chemical kinetics formalism using ordinary differential equations (ODEs) to represent mass-action chemistry.

Biochemical signaling models frequently suffer from incomplete parameterization. Thus 'detailed' models of signaling pathways, which are of course essential for many kinds of mechanistic analyses and design of experiments, are often under-constrained. In this context, a reduced model is preferable as it requires fewer parameters. One frequently used form specifies rate of change of concentration of each molecule as a weighted sum of input molecule concentrations, which may be passed through a sigmoid to achieve saturation [16–18]. This form is quite similar to neural network models. Thus it lends itself to machine learning approaches to obtain parameters from systematic experimental time-series measurements [17]. The authors obtained relatively sparse interaction weight matrices, thus keeping down the number of parameters. While this formulation is effective at modeling dynamics of molecules in reaction networks, the resultant interaction matrices do not map directly to reaction pathways. Similarly, other formal approaches to model reduction yield very compact models, but the mapping to experimental observables may be quite indirect [15]. Hence it is useful to have a compact chemically-inspired formulation to serve as the core for the model reduction while remaining easy to parameterize and predict using the same quantities that are measured in experiments [19–21]. Indeed, a compact model with few parameters is arguably a better starting point to understand complex signaling with insufficient data, than is a mechanistically detailed model.

Savageau and colleagues have developed the Design Space Toolbox to facilitate a systematic approach to developing reduced signaling and transcriptional network models with specified properties such as multistability [22]. They cast mechanistic models into a Generalized Mass Action form, and this is then analyzed to realize the required phenotypic repertoire. While this is an effective way of obtaining models with desired multi-state properties, it differs in objectives from our goal of having a reduced, very efficient representation of dynamic responses of complex reaction networks such as synaptic signaling.

Efficiency is a specific constraint in developing models of synaptic signaling. On the one hand, many neural functions depend on the nuances of signaling. For example, network properties are quite sensitive to different plasticity rules [23], neuromodulators [24], and mutations [25]. Network models also are expanding to include diffusible messengers controlling cellular activity and blood flow [26]. At the single-cell level, explorations of receptor insertion and clustering [27,28], sequence recognition [29] and synaptic tagging [30,31] all require some level of reference to the chemical signaling. The crux of the problem arises when these studies need to scale beyond one synapse to whole-neuron (up to $10^4$ synapses, [29]) or even network scales (e.g., $10^9$ synapses [32]). Clearly, efficiency in memory and computations is important for such models.

The HillTau form addresses several key concerns with modeling of complex signaling networks. It utilizes only those observable states specified by the user to map directly to the chemistry, thus supporting sparse models that are easier to constrain with limited data. This requires very few parameters, yet behaves similarly to chemical cascades involving multiple intervening steps. Since the user specifies their chosen observables, each can be related directly to observations of concentration over time. The models are small and calculations are highly efficient, being closed-form and event-driven.

## Results

We first provide an overview of the HillTau algorithm. Then we illustrate its use to approximate increasingly complex reaction networks. We then show how one can reduce a mass-action model to its HillTau equivalent, with a tradeoff of greater complexity for better accuracy. Finally we carry out some benchmarks of several reduced HillTau models against the original ODE-chemical kinetic models run on two simulators, MOOSE [33] and COPASI [12], and show that HillTau is orders of magnitude faster.

### Overview of HillTau algorithm

The name HillTau comes from combining the Hill form for concentration-dependence of a reaction, and *tau*, the time-course for settling to steady state. In brief, a 'reaction' in HillTau uses the Hill equation with modifiers to estimate steady-state values $Y_\infty$ of the product of one or several chemical reactions having an input reagent $Y_{input}$, and a Ligand $L$, with order $n$:

$$Y_\infty = Y_{input} L^n / (KA^n + L^n) \qquad\qquad \text{Eq i}$$

It may also optionally have a modifier $M$, with order $h$:

$$Y_\infty = \frac{Y_{input} L^n}{L^n + KA^n (1 + (M/K_{mod})^h)/(1 + A_{mod}(M/K_{mod})^h)} \qquad\qquad \text{Eq ii}$$

A modifier changes the effective KA of a reaction, and is controlled by two terms. $K_{mod}$ determines the half-max concentration of the effect of the modifier. $A_{mod}$ determines what

effect the modifier has on the reaction. If $A_{mod} < 1$, the modifier is inhibitory, else it is excitatory [34]. The steepness of the effect of the modifier is controlled by its order, $h$.

The HillTau formulation of a reaction also incorporates $\tau$, the time over which the system exponentially approaches this steady-state. We allow for different time-courses $\tau$ and $\tau_2$ when the concentration is rising or falling:

$$\text{If } Y_\infty > Y(t) \qquad \frac{Y(t + \Delta t) - Y(t)}{Y_\infty - Y(t)} = 1 - exp(-\Delta t/\tau) \qquad \text{Eq iii}$$

$$\text{If } Y_\infty < Y(t) \qquad \frac{Y(t + \Delta t) - Y(t)}{Y_\infty - Y(t)} = 1 - exp(-\Delta t/\tau_2) \qquad \text{Eq iv}$$

This exponential form is a good and efficient approximation to the differential equation form for reaction rates (Eq v), so long as the timestep $\Delta t$ in Eqs iii and iv is smaller than $\tau$ (See methods):

$$Y'(t) = (Y_\infty - Y(t))/\tau \qquad \text{Eq v}$$

The set of elementary HillTau reactions are illustrated in Fig 1, and the details of the calculations are provided in the Methods section.

The motivation for this formalism is that the steady-state value of a cascade of binding reactions, or of enzyme reactions with a fixed rate back-reaction, can be approximated by a Hill function (Methods). Further, the time-course of approach to steady-state is typically governed by the slowest reaction, and this can be approximated as an exponential settling function (Methods).

Note that we do not assume that the input, activator and modifier act in a single mass-action chemical step. Indeed, HillTau is most effective for model reduction when one can fit several mass-action steps using one HillTau 'reaction'.

Since Eqs i to iv are analytic, one can do this calculation in an event-driven manner. HillTau achieves sparseness and simplicity by approximating many steps with a single 'reaction', considering only those intermediates that are needed for readouts or for improved precision. It achieves speed because the models are smaller, and by using event-driven calculations rather than numerical integration.

Most reaction networks cascade through many layers of reactions. HillTau evaluates each upstream layer before downstream ones. It first builds a dependency graph of all reactions. This is done by identifying input molecules as layer 0, and successively ranking all reactions that depend only on layer 0 as layer 1, reactions that depend on layers 0 to 1 as layer 2 and so on.

HillTau identifies feedback loops by reactions which do not resolve into the above layers. Based on ordering of reactions in the model definition, it picks a reaction to 'break' the loop, and assigns it to layer N+1, where N was the previously deepest layer. It then repeats the process of layer assignment, including further loop-breaking if needed.

During evaluation of a single step in HillTau, all the steady-state and time-course calculations are completed for layer 1, then layer 2 is calculated, and so on. Thus each layer receives the inputs appropriate to the current time before doing its evaluation. In cases where update events are separated by periods greater than the shortest $\tau$ in the system, additional time-steps are inserted to maintain accuracy (Methods). For typical use-cases, such as synaptic plasticity models, the event interval is shorter than the time-courses in the model (typically ~1 sec) and hence only a single step is taken. In cases where HillTau inserts additional time-steps for accuracy, it is done behind the scenes of the same event-driven programming interface. If there is

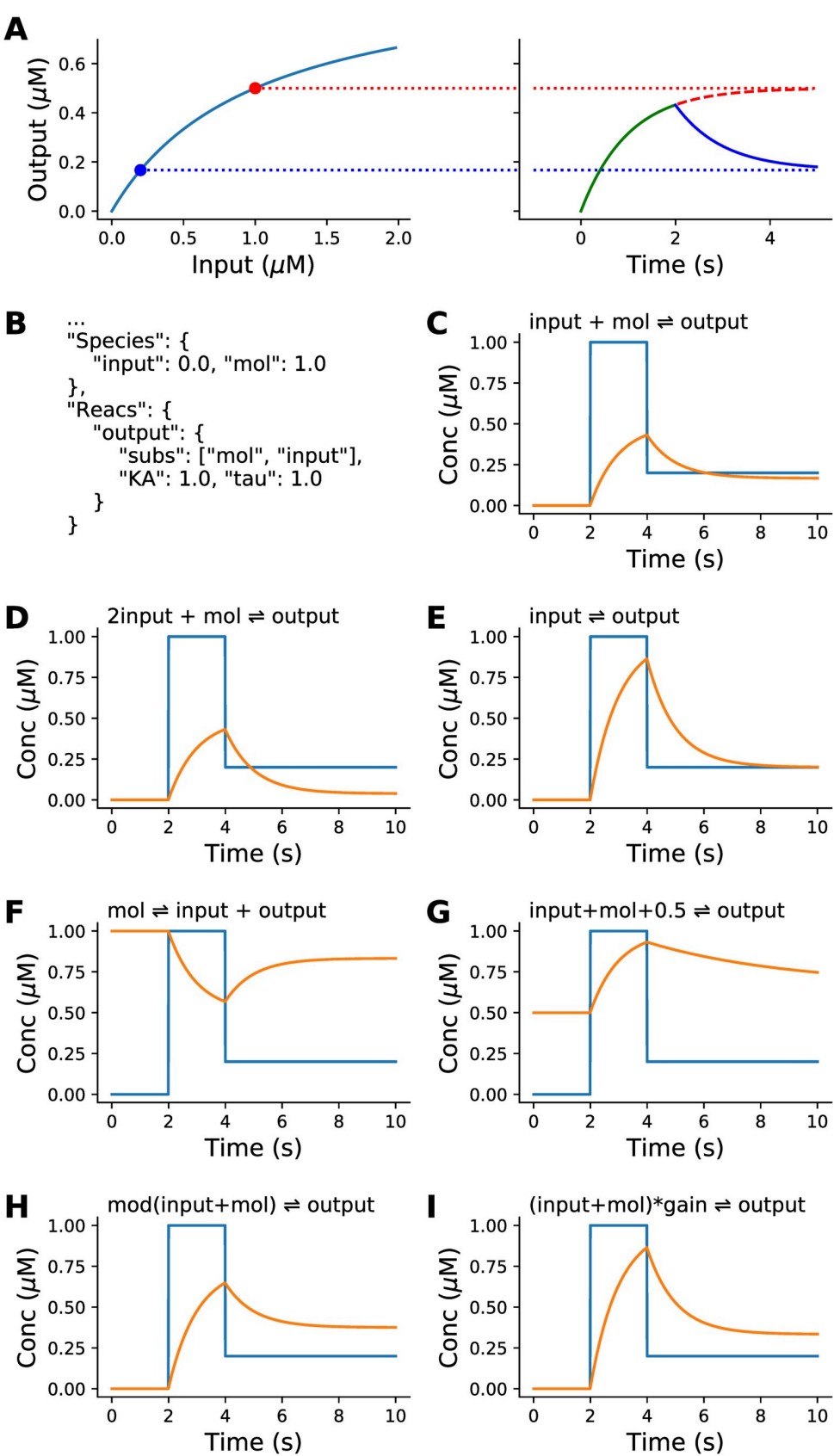

**Fig 1. The HillTau formulation and representation of elementary chemical reactions by a single HillTau reaction.**
A: Principle of HillTau formulation. Left: steady-state output values for different levels of the input molecule,
computed by the Hill equation. In all simulations in this figure, two input values are used: first 1 μM (red dot) and later
0.2 μM (blue dot). Right: The simulator starts from the current value of the output, and computes the approach to the
steady-state as an exponential time-course. Note that these are algebraic calculations, not numerical integration. In this
example the output rises from zero toward the red dotted line for 2 seconds. Then the input is changed to 0.2 μM, and
now the simulator approaches the steady-state value for this (blue dotted line) with an exponential time-course. B: Key
section of the JSON code defining this reaction system. C-G: Inputs (blue) and simulated time-course of outputs
(orange) for seven different reactions. Each is represented by a similar HillTau reaction but with different parameters
(see S1 Data). In all cases the HillTau output onset is identical to the output computed using numerical integration of a
single reaction expressed as mass-action chemical kinetics. Decay time-course may differ from onset time-course in
mass-action. C: Binding. D: 2nd order binding. E: Conversion. F: Inhibition, conceptually equivalent to removal of
output molecules by binding of input to the output molecule, and sequestration of the resultant complex. G: Variant of
binding reaction, in which there is a fixed baseline of 0.5 μM, and the system has different on (tau = 1s) and off
(tau2 = 5s) time courses. H: Same as reaction 1, but with a modifier term that strengthens input affinity. I: Same as
reaction 1, but with a gain term that multiplies the output, in this case by a factor of 2.

feedback, then again one has to use event intervals shorter than the shortest τ in the feedback
loop. A factor of 10 usually gives good convergence (e.g., S7A and S7D Fig).

In summary, HillTau uses analytic evaluation of reaction outputs based on a Hill-like form
and exponential settling, and propagates the evaluation through successive layers of the reac-
tion network for each event time. Events can be stimuli or points in a time-series for sampling
system time-evolution.

## The HillTau form can model a range of chemical signaling motifs

We implemented a range of elementary chemical signaling functions to illustrate the use of the
HillTau form (Methods, Fig 1). The HillTau versions of most of these reactions have an exact
fit to their mass-action counterparts (S1 Fig). We further implemented key signaling motifs,
including feedback inhibition, oscillation, and bistables (Fig 2). To do this, we constructed
minimal HillTau schemes that incorporated the essential elements of each of these motifs. We
developed an optimizer program *mash.py* (MASH: Model Abstraction from SBML to HillTau,
see Methods) to tune parameters of the HillTau models to match the outputs to the original
mass-action or ODE versions. MASH runs the reference model through a range of stimuli
designed to explore its input-output properties, and then uses numerical optimization meth-
ods from scipy.optimize to tune parameters so that the HillTau model produces a good fit to
the original. We used normalized RMS difference between the traces as a measure of goodness
of fit. In Fig 2A–2C we compare feedback inhibition implemented in mass-action (5 reactions,
7 species, 2A), HillTau (2 reactions, 3 species, 2B), and run for a square pulse input (2C). The
feedback inhibition model is well approximated by the HillTau version to within 4% normal-
ized RMS deviation.

Next, we implemented a HillTau version of a mitogen-activated protein kinase (MAPK)
feedback oscillation model having 11 reactions and 15 species, Fig 2E [35]. We used three Hill-
Tau reactions to map to the key components of the original ODE model. First, we used a reac-
tion to represent the basic MAPK cascade. Second, we provided an output reaction to
represent the phosphorylation of the MAPK molecule by the cascade. While it was possible to
use this output signal to inhibit the cascade, we found we had to implement a separate reaction
for the negative feedback step to introduce a longer delay to match the observed oscillations.

Having constructed the model structure, we next fit the HillTau model to the original ODE
model using MASH. As initial parameter estimates, we used taus of the order of the oscillatory
period, and KA of the same order as the (known) molecular concentrations. We first fit the ini-
tial output transients. Then we ran it for a complete cycle. Finally we stretched the fit time to

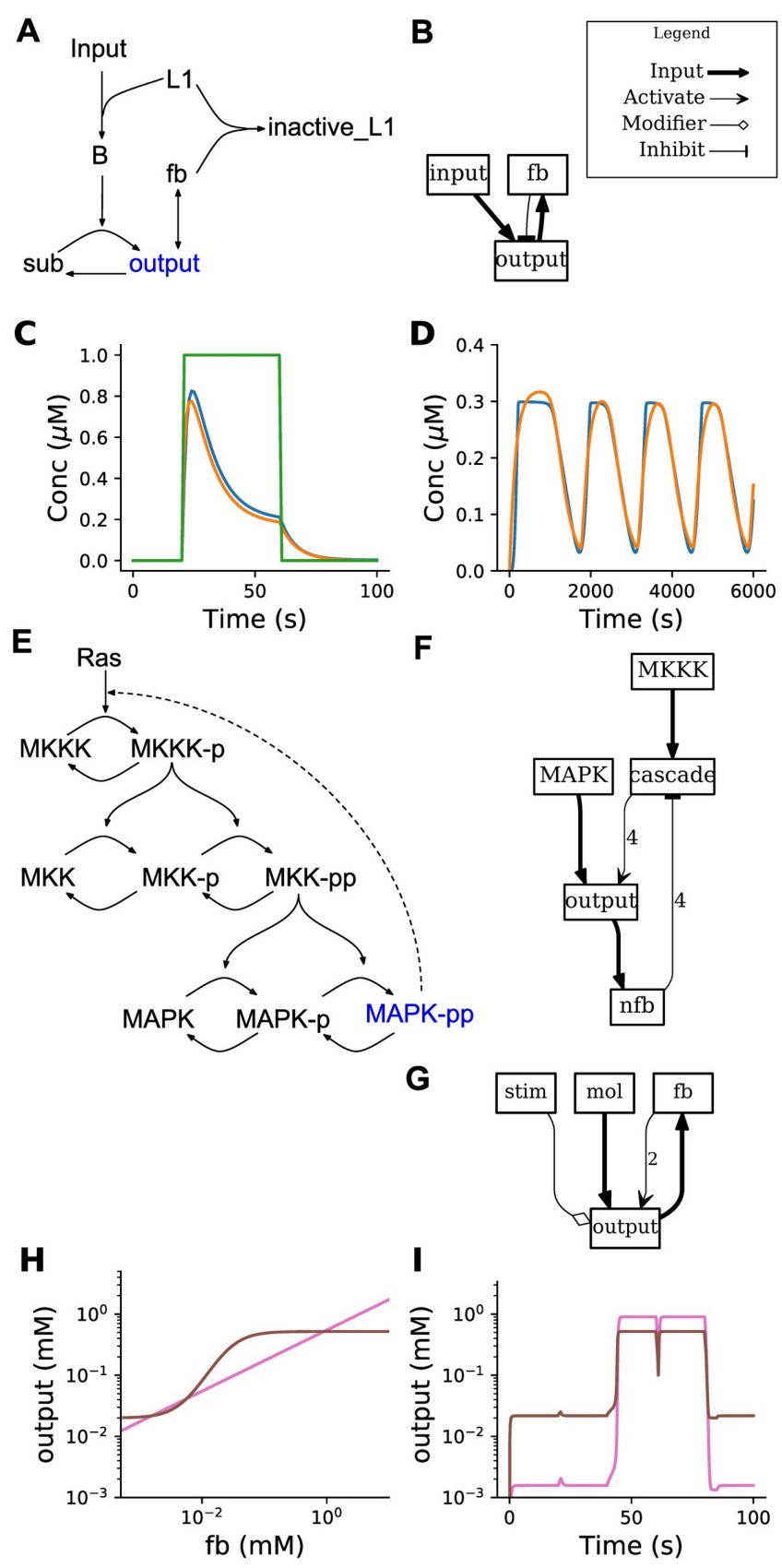

**Fig 2. HillTau models of key signaling motifs.** A-C: Feedback inhibition. A: Mass-action reaction scheme for feedback inhibition, involving 7 molecules and 5 reactions. B: HillTau version. Each box represents a molecule. If there are input arrows to the box it means there is a reaction whose product is the named molecule. Input arrows can be either inputs (reagents), activators, inhibitors, or modifiers. This reaction consists of 3 molecules (input, fb, and output) and 2 reactions (fb and output). C: Simulations for mass-action (blue) andHillTau (orange) versions of feedback inhibition. The green trace is the input molecule. D-F: Oscillator from ultrasensitive MAPK cascade, taken from [35]. D: Output of simulation. Blue is ODE output and orange is HillTau. E: ODE model. This uses 15 molecules, and 11 reactions. MAPK-pp is the molecular species used as output of the oscillator. F: HillTau reaction scheme for oscillator, using 5 molecules and 3 reactions. The concentration of the 'output' molecule is plotted. G: HillTau model of bistable system, involving 4 molecules and 2 reactions. H: Phase plot showing stable states of system as the intersection points between the steady-state dose-response curves. This was generated by varying the feedback molecule *fb*, and measuring *output* (brown curve), and then varying the *output* molecule and measuring *fb* (pink curve). I: Time-series illustration of state switching in the bistable. As before, *output* is in brown and *fb* in pink. The Y axes of H and I are the same to show that the steady-state output levels (brown) match. The system starts in the low state. At 20 s a small excitatory input *stim* is given which fails to switch the state. At 40 s a strong input causes switching to the high state. At 60 s a weak inhibitory input fails to turn it off, but at 80 s a strong inhibitory input returns the state to baseline. Excitatory and inhibitory inputs were delivered by transiently setting the level of *stim* to high or low values.

include a few cycles. This incremental increase in time was necessary because our RMS scoring function gives very poor scores for otherwise good models if a phase mismatch builds up over a few cycles. The final reduced HillTau version (3 reactions, 4 species, Fig 2F) had similar period and amplitude (Fig 2D), and it fit the waveform to within ~7.6% normalized RMS deviation.

Finally, we made a HillTau version of a chemical bistable switch using just 2 reactions (Fig 2G). We demonstrated that the HillTau form works with the standard dose-response (null-cline) approach to estimating steady states (Fig 2H) and showed that the resulting switch exhibits high and low states that are triggered by transient inputs (Fig 2I).

Thus the HillTau form can efficiently represent a range of important signaling motifs and their dynamics, including feedback inhibition. oscillations, and bistability.

## The HillTau form compactly represents bidirectional synaptic plasticity

Synaptic plasticity is one of the most-modeled neuronal signaling processes [4,36]. The key features that have been represented include stimulus strength-dependence, timing dependence, and long-term state storage [37]. A few studies have come up with rather detailed models to implement each of these processes [31,38,39]. As an illustration of all these properties in the HillTau system, we implemented bidirectional synaptic plasticity including long-term synaptic state changes (Fig 3). One of the interesting aspects of synaptic plasticity is that in many systems, the same input modality (typically read out as $Ca^{2+}$ concentration) can give rise to both synaptic depression and potentiation. This has significant theoretical implications and an abstract rule for this bidirectional plasticity was proposed by Bienenstock et al. (the BCM rule, [40]). We first devised a simplified mass-action version of the BCM rule using 9 molecules and 6 reactions (Fig 3A). The species p_AMPAR is the phosphorylated form of the receptor, assumed to be inserted into the synapse. Here, resting $Ca^{2+}$ does not alter the state of the model; low $Ca^{2+}$ causes depotentiation (that is, reduction of receptor levels), and high $Ca^{2+}$ causes potentiation (Fig 3C, 3D and 3E). We then implemented a BCM model in just 3 reactions in HillTau (Fig 3B). We used the program *mash.py* to fit the HillTau model to the reference mass-action version using a set of generic time-series and dose-response stimuli (Methods). We obtained a normalized RMS fit of ~2.3%. When we used the fitted model for Fig 3, we obtained fits of ~5.2%, 8.9% and 2.3% for panels C, D and E respectively even though the model had not been tuned to these stimuli. As a further elaboration, we introduced a bistable switch for long-term retention of synapse state, which was driven bidirectionally by the BCM rule (Fig 3F). The bistable switch, derived from Calcium-calmodulin Type II kinase

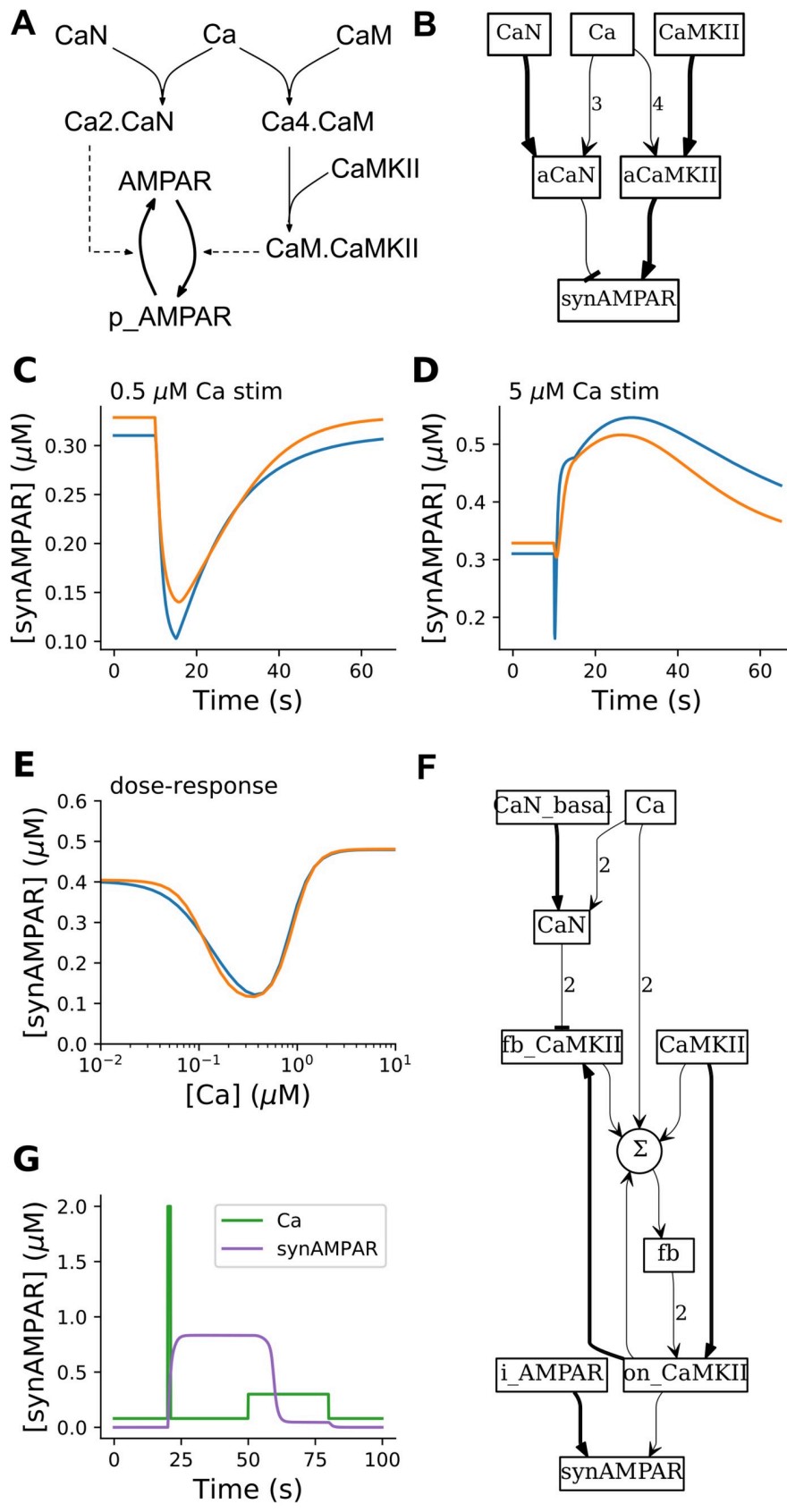

**Fig 3. HillTau version of synaptic plasticity rules.** A. Mass-action model for generating Bienenstock-Cooper-Munro (BCM) rule for synaptic plasticity. p_AMPAR is the phospho-receptor, and is the output of the model. It is assumed to localize to the synapse and is thus also referred to as synAMPAR. Calcium triggers both an inhibitor (Calcineurin, CaN) and a stimulus (CaMKII) for receptor phosphorylation and insertion into the synapse. CaN activates at lower $[Ca^{2+}]$, so there is initially a reduction in p_AMPAR. CaMKII is present at very high levels, so at higher $Ca^{2+}$ it out-competes CaN to give an increase in p_AMPAR. B. BCM rule implemented in HillTau. Here the species synAMPAR is the output of the model. C-E: Comparison of mass-action model p_AMPAR with HillTau model synAMPAR. Orange is HillTau, blue is mass action. C: 1s stimulus at 0.5 μM $Ca^{2+}$ gives a reduction in synaptically localized AMPAR (synAMPAR). D: 1s stimulus at 5 μM $Ca^{2+}$ gives an increase in synAMPAR. E: Dose-response curve of steady-state synAMPAR as a function of $[Ca^{2+}]$ for mass-action (blue) and HillTau (orange) models. In both cases settling time for each point was 1000s. F: Schematic for BCM rule model feeding into bistable model, implemented in HillTau. The circular node labeled Σ represents weighted summation of multiple inputs. G: time-course of simulation of bidirectional plasticity using different $Ca^{2+}$ stimuli. At t = 20, a 1s stimulus of 2μM $Ca^{2+}$ (green trace) causes a transition to the active state, using synaptic AMPAR (maroon trace) as a readout. At t = 50s, a 30s stimulus of 0.3 μM $Ca^{2+}$ pulls the system back to resting state.

(CaMKII) signaling, controls receptor insertion. Using this model we delivered a typical potentiating stimulus (strong but brief $Ca^{2+}$ input), leading to sustained synaptic AMPAR elevation. We followed this with a typical long-term depression stimulus (modest but sustained $Ca^{2+}$ input), which turned the switch off again and led to reduction in AMPAR (Fig 3G). This composite model required 4 reactions and one summation function in the HillTau form. Several mass-action models of synaptic state switches include these elements (e.g., [4,36,38,41]) and they typically involve far more molecules and reactions (e.g., the Hayer and Bhalla 2005 model used 133 molecules and 215 reactions [38]).

Overall, these examples illustrate how compact HillTau models can represent both the bidirectional induction of plasticity, and also long-term maintenance of synaptic state.

## HillTau models can be optimized to fit biochemical measurements

The above examples illustrate how HillTau can represent biological signaling motifs, and build them up into networks with interesting computational properties. We next approached a complementary problem in signaling, namely, to take a complex signaling system, and fit simple HillTau models to it. This provides a way to perform model reduction and to infer computational properties. The basic flowchart is illustrated in Fig 4. This flowchart addresses both the heuristics of defining model topology, and of parameter fitting.

The heuristics for defining model topology are as follows.

1. Identify inputs and key readout molecules. These readouts may be important (and experimentally measured) intermediate signaling molecules in a reaction network, or the end-products of a cascade.

2. Assign a reaction for each readout molecule, with an input as an upstream substrate or inactive state of the molecule, an activator (or inhibitor, see methods) and optionally, a modifier. Together these control the level of the readout molecule.

3. In case a molecule has multiple inputs, bring in additional reaction steps based on the known reaction mechanisms. For example, if we have BDNF, EGF and Ca all controlling ERKII activity, then we could specify an intermediate step where the two receptor tyrosine kinase ligands converge, and this combination is an activator for the ERKII reaction with Ca as a modifier.

4. In case a readout is simply the sum of multiple active states of a molecule, use an equation to define this summation.

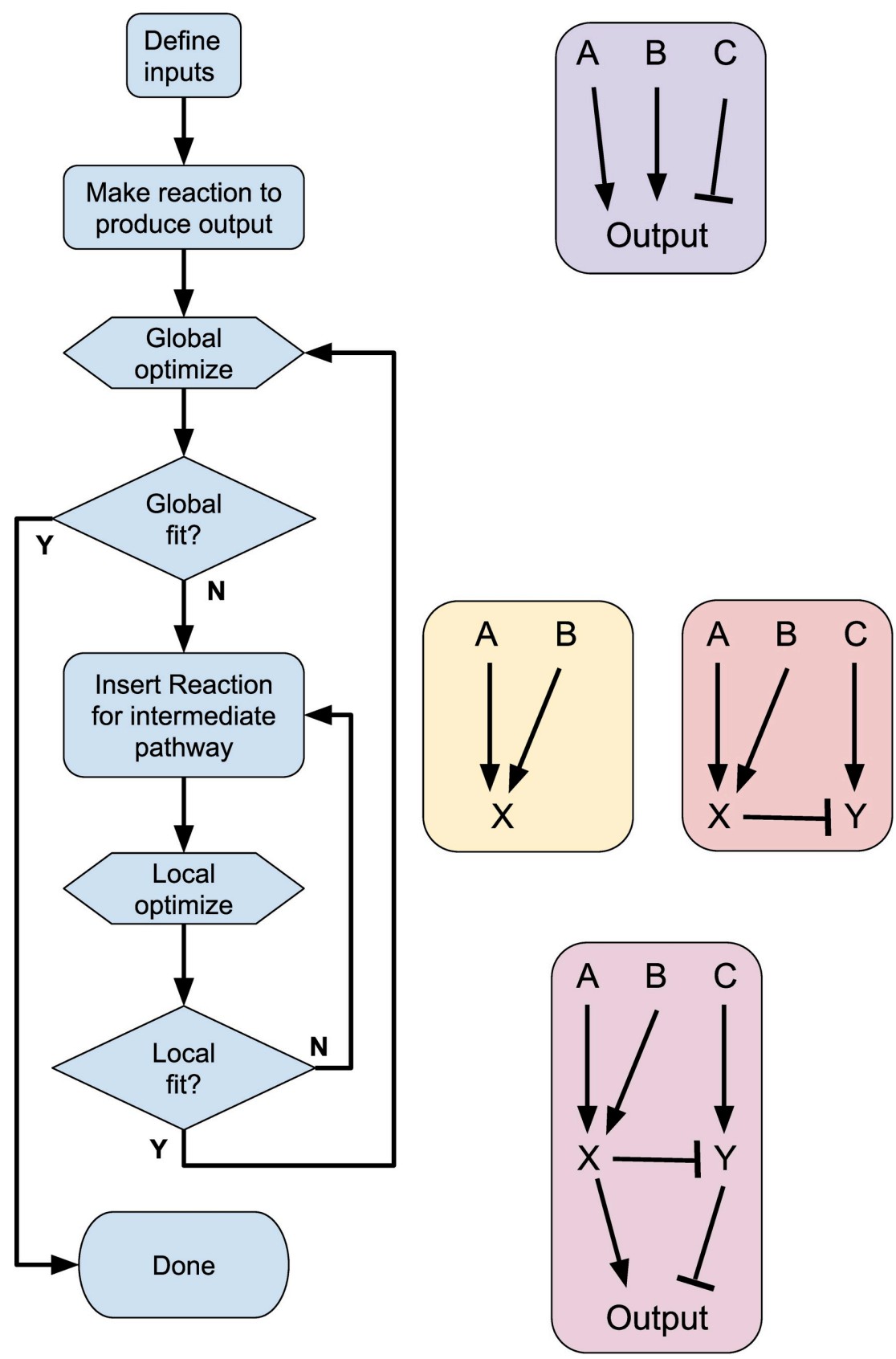

**Fig 4. Flowchart for model building using HillTau.** Left: flowchart. Right top: initial reaction with inputs A, B, and C. Right middle: successive local increments to model, introducing reactions X and Y respectively. Right lower: Final HillTau scheme with good fit at all stages.

5. Obtain best model fit as per flowchart in Fig 4. If model accuracy does not meet criteria for your objectives, identify poorly performing intermediate readouts and insert further intermediate reaction steps.

Note that in principle many of these steps can be automated. For example, one can generate a family of models algorithmically (e.g., [42]) and optimize over topology as well as parameters, but this is out of the scope of the current study. Other algorithmic approaches for model reduction are discussed in [14].

In the current paper we have used the above heuristics to generate HillTau schemes (model topology) by hand.

There are also simple steps to obtain initial parameter estimates for each HillTau 'reaction':

1. KA from the activator concentration at half-maximum of the experimental activation curve, or directly from mass-action model rates.

2. Time-course $\tau$ from the experimental time-course of the reaction, or from the slowest intermediate step of a mass-action model. If there is a distinct time-course when the reaction turns off, use this as $\tau_2$.

3. When the modulator is present, assign Kmod and Amod from the half-maximum and the steepness of modulation curve.

This approach works in the same way for model construction from experimental response curves, and for model reduction from response curves taken from detailed models. Following generation of an initial, roughly parameterized HillTau model, we can deploy the model fitting approaches described in FindSim[43]. In brief, FindSim provides a Python-based framework for matching models to experiments. It codifies the experiment design (e.g., time-series, dose-response, bar-chart) and experimental results into a single machine-readable file. FindSim runs the experiment on the model and returns a numerical score for goodness of fit. The model may be defined in SBML (run using MOOSE) or using HillTau. Thus FindSim can be used as the scoring function for optimizing model fit to experiments using a variety of optimization methods available in scipy.optimize.

For the special case of model reduction, where we already have a detailed SBML model and wish to fit a reduced, HillTau version, the utility MASH provides a shortcut alternative to the FindSim and optimization pipeline (discussed above and in the Methods section).

As an example of this flowchart and the use of HillTau fitting to match an existing, detailed chemical ODE model, we derived a HillTau model of synaptic activity-triggered protein synthesis. Our reference data was obtained by running a series of 'experiments' on a published model implemented in mass-action kinetics [44]. The original model was based on numerous experiments, and included 123 molecules and 120 reactions (Fig 5A). The input pathways were $Ca^{2+}$ and brain-derived neurotrophic factor (BDNF), and the final output was protein synthesis rate.

We started with the most reduced form, a single reaction to replace the entire synaptic protein synthesis network. We specified amino acids as the input, BDNF as an activator, Ca as a modifier, and protein as the product of this reaction, (Fig 5B) to obtain our starting reduced model. We used MASH to carry out the optimization (Methods). In a model with a single reaction, MASH obtained a fit of about 11% normalized RMS. This is remarkable for such a simple

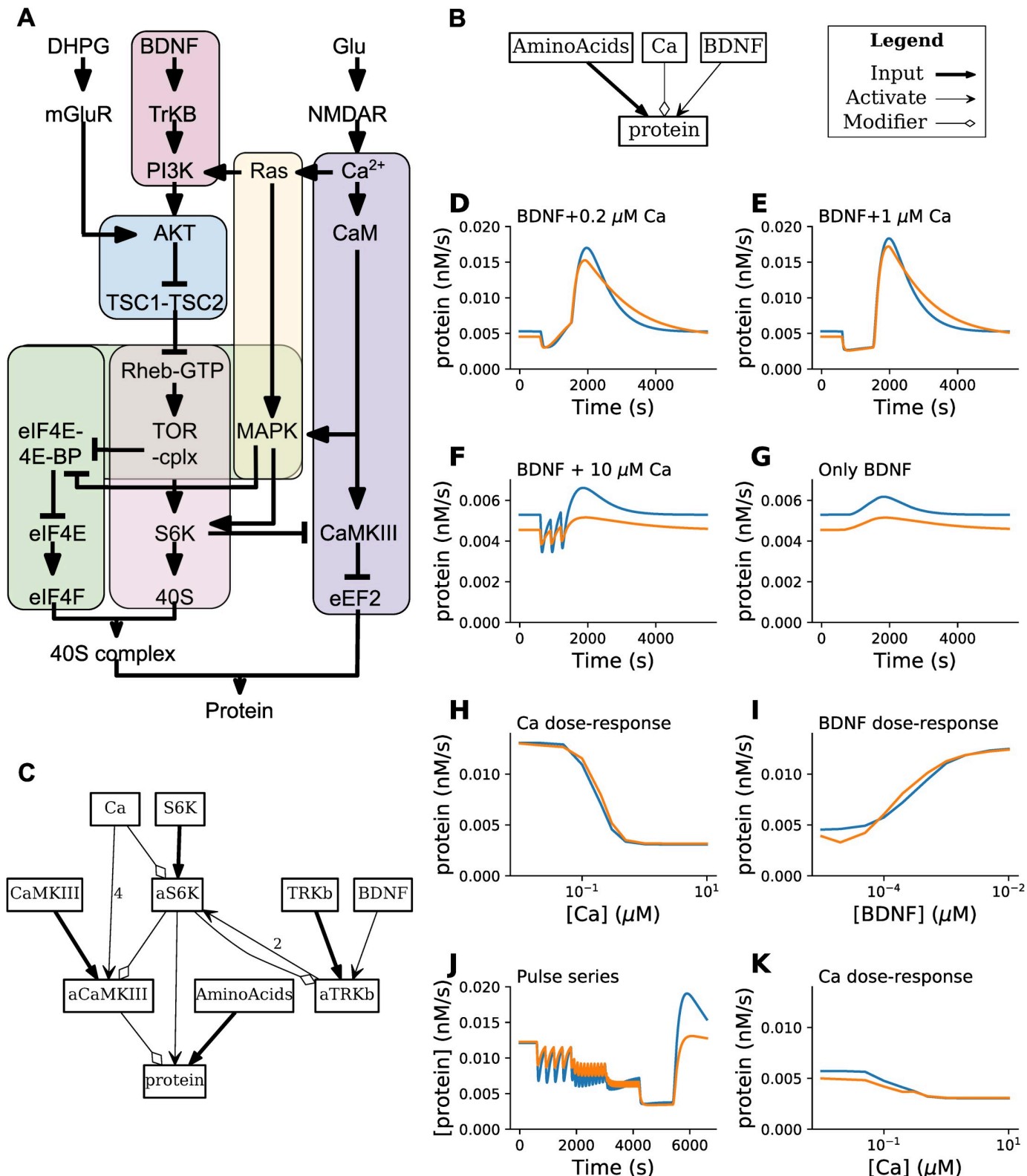

**Fig 5. Model fitting and model reduction.** A: Block diagram of source model with 123 molecules and 120 reactions, from [44]. B: First pass reduced model in HillTau, with 1 reaction and 4 molecules. C: Final reduced HillTau model including activated S6K (aS6K) and activated CaMKIII (aCaMKIII) as intermediate readouts which also were fit to data. This model has 4 reactions and 10 molecules. D-K: Eight 'experiments' on reference and HillTau models, not part of stimulus

set used to tune parameters for HillTau version. In all cases protein production rate is readout. Blue plots are reference, orange areHillTau. D: BDNF@3.7 nM + Ca$^{2+}$@0.2 μM, 900 seconds. E: BDNF@3.7nM, Ca$^{2+}$@1μM. F: 3 pulses of BDNF@3.7 nM for 5s, coincident with Ca$^{2+}$@10μM for 1s, pulses separated by 300 s. G: Same as F, but Ca$^{2+}$ held at baseline of 0.08 μM. H: Dose-response of protein vs. Ca$^{2+}$, holding BDNF fixed at 3.7 nM. I: Dose-response of protein vs BDNF, holding Ca$^{2+}$ fixed at 0.08 μM. J: Protein production rate for fixed BDNF@3.7 nM, where Ca$^{2+}$ was given in 1 second pulses at intervals of 300, 120, 60 and 10 seconds; each pulse sequence lasting for 1200 s.. K: Dose response of protein production rate vs. Ca$^{2+}$, holding BDNF at basal levels of 0.05 nM. The average normalized RMS difference across the eight 'experiments' was under 10%, and in all cases the qualitative properties such as direction of change, matched well.

model. It does well with the dose-response experiments (Scores of 7% and 5%, S2 Fig). However, it does not do a good job of replicating the dynamics of the experimental data, achieving scores in the range of 10% to 42%. (S2 Fig). We therefore increased the HillTau model detail. To do this, we introduced two additional key intermediates into the HillTau model: S6K, and CaMKIII. We first made a HillTau model involving inputs to S6K alone. Based on the known pathway, we chose BDNF as the activator, and Ca as the modulator for S6K. The S6K responses to combinations of these two inputs were used in MASH to obtain a fit to within 3.3% (Individual panel fits were between 3% and 22%, see S3 Fig). We then held S6K parameters fixed while we fit CaMKIII. CaMKIII is activated by Ca, and modulated by S6K. MASH gave a CaMKIII fit of 1.9% (Individual panels 2% to 20% but most of the poor scores were small differences at baseline; S4 Fig).

Collectively, the optimizations for S6K and CaMKIII correspond to the inner loop of Fig 4. Finally, we added a final protein synthesis reaction that took the already fitted S6K and CaMKIII activity as activator and modifier. We held the earlier reactions (S6K and CaMKIII) fixed, and optimized only the protein synthesis reaction. After this, the composite model fit the optimization waveform in MASH to within 3.2% (normalized RMS), and the figure panels fit within a mean of 9.3% (Fig 5D–5K).

At this stage one could choose to perform further optimization in a couple of ways. We could have obtained closer fits had we optimized to the same stimuli as in the figure panels. Instead we used more general input time-series and dose-response stimuli to the MASH optimizer to see how well the model would generalize. This gives a HillTau model that behaves well across a wider range of conditions than the experiments in Fig 5. Second, there is a small systematic difference in baseline in panels 5F and 5G arising from a difference in output at resting BDNF, seen in panel 5I. The introduction of additional intermediate reactions in the model as per Fig 4 could further improve the fit. Such tradeoffs between generality, accuracy, and model complexity are common, and given that HillTau is meant for building compact models we considered this fit sufficiently good for illustration.

Overall, the abstract HillTau model captures many of the key properties of the mass-action system. These include steady-state and time-series responses of two inputs (BDNF and Ca), and three readouts: S6K, CaMKIII, and the end-product protein (Figs S3, S4, and S5 respectively). This is a highly effective dimension reduction, from over 360 to 31 parameters.

It is important to note that the efficiency of HillTau made the optimization calculations quite tractable. In the final optimization run for the entire model, the single reference ODE run took ~60 seconds, and the cumulative time for 746 HillTau evaluations was around 18 seconds. The optimization algorithm itself took about 75 seconds, excluding function evaluations.

Can we create a reverse mapping from these simplified HillTau models to ODE forms? A close but not exact mapping is obtained by taking the small-time limit of the HillTau event-driven form (Equations 3.x) and converting to an ODE (rate) form (Equations 4.x). ODE equations are supported by many systems biology simulators. It is not an exact mapping because HillTau may use different values for rising and falling time-courses (tau and tau2), whereas the ODE form can accommodate just a single value, tau. We implemented this conversion in a program, *ht2sbml.py*, which is provided on the GitHub repository for HillTau.

Using this mapping we were able to export HillTau to SBML, and tested that SBML-capable simulators such as COPASI could run the reduced, ODE form models, and give approximately matching results (Methods, S5 Fig). Thus we can use the HillTau toolchain to make reasonably reduced ODE models, though these are neither as efficient as their HillTau counterparts, nor do they have the same capabilities to use two time-courses to improve model fitting.

In summary, we developed a systematic procedure for developing reduced HillTau models to fit mass-action simulations, including a model optimization utility MASH. We illustrate this procedure by developing a HillTau model of 10 molecules and 4 reactions to fit a mass-action model having 123 molecules and 120 reactions. The resultant HillTau models generalize well and the fit improves when intermediate reaction steps are added.

## HillTau models are compact and efficient

We next took a set of HillTau models of various levels of complexity, and compared various measures of computational cost with the ODE equivalents (Table 1 and Fig 6.)

We first compared model complexity, measured as the number of parameters needed to specify the model. The number of parameters scales roughly as

*# of molecular species + 2 * # of reactions.*

This is a slight overestimate, since some of the molecules are state variables and we do not need initial concentration values for them. Here we consider state variables to be those which are computed, as opposed to defined using initial conditions. In ODE models we estimate this by counting the number of rate terms plus the number of molecular species with a non-zero initial value. In HillTau models we count the rate terms and the species which are not reaction outputs. This yields the approximate scaling terms below. Each reaction needs two parameters, *Kf* and *Kb* for conversion reactions, and *Km* and *kcat* for enzymes. We sampled from among the mass-action models presented in the above sections, ranging from 3 to over 360 parameters, and included an additional model with almost 750 parameters. (Fig 6A). The HillTau form had a similar scaling with molecules and reactions, except that HillTau also allows for a number of optional terms in the reactions so the average scaling is somewhat larger than *2 * # of reactions*. We found that the HillTau form became increasingly effective at model reduction for larger models. Note that here the optimization goal was to obtain a single end-point response (3 end points in the case of the model in Fig 5). Further reactions would be needed to also represent intermediate pathway readouts.

**Table 1. Parameters used to define a HillTau model.** Concentration units can be any of M, mM, uM, and nM, and are specified in the JSON file. The default concentration units are mM.

| Parameter | Units | Meaning | Default | Required? |
|---|---|---|---|---|
| ConcInit | Concentration (can be any of M, mM, uM, nM) | Initial concentration. Only applies to species definitions | 0 | No |
| KA | Concentration | Association constant | N/A | Yes |
| τ | Time (seconds) | Time course for relaxation to steady state | N/A | Yes |
| τ2 | Time (seconds) | Time course of relaxation if output is falling. | τ | No |
| Gain | None | Scaling factor for reaction output. Used to indicate enzymatic amplification. | 1 | No |
| Baseline | Concentration | Baseline value of reaction output | 0 | No |
| Kmod | Concentration | Half-saturation concentration for modifier | N/A | Only if modifier molecule is specified. |
| Amod | None | Activation term for modifier | 4 | No |
| Nmod | None | Order of modifier action | 1 | No |

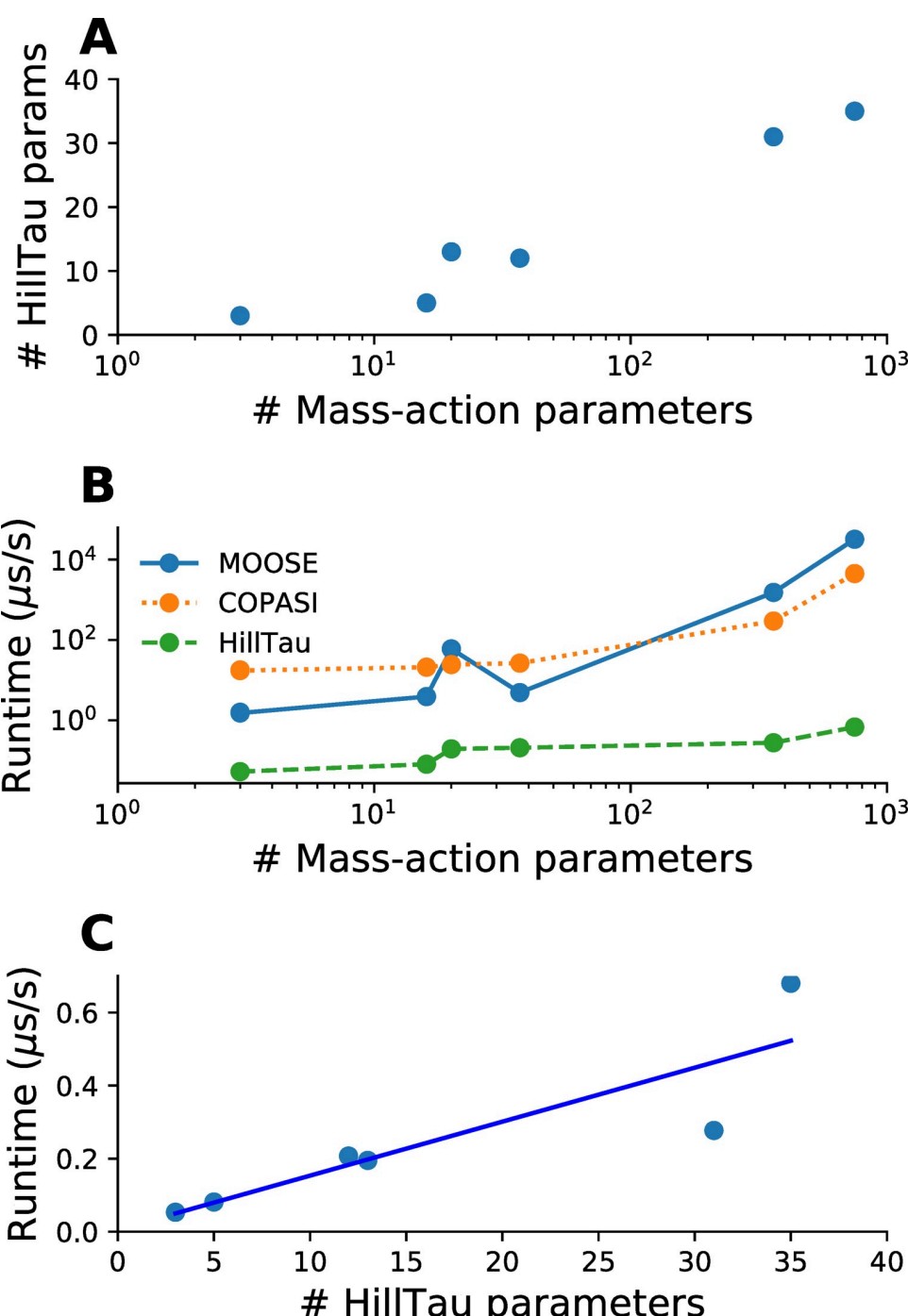

**Fig 6. Efficiency of HillTau method.** A. Scaling of number of parameters. The HillTau form uses far fewer parameters than mass-action, and becomes relatively more concise with larger models. B, C: run-time scaling. All run-times are expressed in μs of wall-clock time to execute 1 second of simulation time. Calculations were done on an Intel(R) i7-7700HQ processor, running Ubuntu 18.04. B. Comparison of run-times of the HillTau and mass-action forms, where the numerically intensive sections of HillTau were implemented in C++, and C++ was also used for ODE calculations in two simulators: MOOSE and COPASI. Due to the combination of model reduction and efficient calculation, HillTau has a huge efficiency advantage which grows to over 3 orders of magnitude for larger models. The accuracy with which this set of HillTau models fit their ODE counterparts was in the range 3 to 9%. C. HillTau model run-time increases with number of parameters. $R^2 = 0.79$, slope = 0.015, intercept = 0.006, units in μs/s.

We then examined run-time efficiency. We took run-times for two ODE simulators, MOOSE and COPASI, whose numerical cores are in C++. We compared these with matching HillTau models run using the C++ version of HillTau (Methods, Fig 6B). For small models, HILLTAU was typically about 100 times faster than the ODE calculations, but for large models HillTau became over 3 to 4 orders of magnitude faster (Fig 6B). This set of HillTau models fit their ODE counterparts within 3 to 9% accuracy, but note that the approximations were inherent in the HillTau model structure, and did not arise from lack of numerical convergence. The run-time for HillTau models grew with the number of parameters (Fig 6C, slope = 0.015 μs/s, $R^2$ = 0.79), but also had a dependence on model stiffness due to the requirement that the internal timestep should be smaller than the smallest τ in the model. This suggests that the HillTau calculation cost could be further improved by utilization of a variable-timestep similar to methods for ODE solutions frequently used for chemical kinetic calculations.

Dose-response experiments are particularly efficient to compute using HillTau. An inefficient way to do these for ODE models is to run them out to steady-state for each successive dose. This may take a long while especially if the system is stiff or converges slowly. It is also possible to use linear algebraic root-finding to find the steady-state value in one step, possibly following a short presimulation to bring the system closer to the steady-state [45,46]. In HillTau, the form itself incorporates the steady-state value, so in principle one could leap to the steady value in one step. To be more conservative, the HillTau does so in 10 steps to smooth out transients and to allow any feedback signals to propagate through the system. As an illustration, COPASI performed the steady-state calculation for a large model (accession 92, DOQCS, ~100 reactions, [44]) in ~1 second after some relaxation of convergence criteria. The HillTau equivalent model (same as final model in Fig 5) took about 4 microseconds.

Overall, HillTau models are compact and highly efficient compared to ODE-solved mass-action models. The efficiency improves for larger models.

## Discussion

We have designed HillTau, a compact, computationally efficient abstraction of chemical signaling that is particularly effective in building reduced models of complex signaling networks. It uses an event-driven algebraic representation based on the Hill equation and exponential relaxation to steady state. HillTau is effective in representing a range of chemical signaling motifs and complex synaptic models, using biological observables of molecules, reactions, association constants and time-courses. We show its applicability for model reduction by optimizing the fit of its responses to those of a reference mass-action model. This generates very compact models. A similar optimization approach works to build a HillTau model directly from experimental data. ThusHillTau addresses many of the concerns of model-building with limited data, and serves as a scaffold for eventual development of more detailed models.

HillTau is phenomenological and semi-heuristic, in that it uses the Hill equation to achieve concentration dependencies that fit well, but ignores many intervening chemical steps. This combination gives it the strong points indicated above, namely speed, compactness, and consistent mapping to experimental observables, but it also sets out clear limitations. Foremost among these is that it can only make limited predictions on detailed pathway chemistry, since it is missing many reaction steps. For example, a HillTau model would be limited in its ability to predict drug targets or side-effects because it may have lumped together potential molecular targets into a single reaction step. It is, however, quite effective in representing and predicting emergent signaling properties because it captures dynamics and topology of signaling networks.

The current HillTau formulation is limited in its handling of two important aspects of signaling in neurons: stochasticity and diffusion. These phenomena are out of the scope of our

current implementation, which has focused on development, validation, simplicity and speed. Many biochemical signaling processes experience substantial stochasticity, particularly in small-volume systems such as the synapse which is a target of our modeling. One possible way to introduce stochasticity would be through the linear noise approximation of the chemical Langevin equation [47], which if used in an event-driven manner could be quite efficient. We anticipate it will take extensive validation to establish its utility in the HillTau framework. Similarly, there are potential ways to elaborate upon HillTau to use an event-driven approximation to diffusion, but these will require later follow-up.

Based on these attributes, we discuss four major use-cases for HillTau: model reduction, system abstraction, scaffolds for data-driven optimization, and efficient approximations to complex cellular signaling.

## Model reduction

Several algorithmic approaches have been brought to bear on the *model reduction* problem, including collapsing multiple mass-action steps into one [48], and power-law generalizations of mass-action signaling [16]. Simulators such as COPASI provide utilities for partitioning a reaction system into fast and slow reactions, thus allowing one to approximate the fast steps with their algebraic counterparts [12,13]. In our test model (Accession 92, DOQCS, [44]) we were able to partition about 80% of the steps into the fast domain (<100 s) using the ILDM method implemented in COPASI. With HillTau one can use a well-known heuristic/optimization approach to simplifying large networks [49,50]. This reduced the same test model down to 4 reactions, about 25-fold (Figs 4 and 5). The approach reported in our current study relies on the modeler starting from a minimal input-> output mapping, then iteratively picking relevant major nodes in the network, and optimizing each subset of the model to fit the data (Fig 4).

Thus one can converge on the minimal set of intermediate nodes (illustrated in Figs 4, 5, and S2, S3, S4) to achieve the desired accuracy of model fit to data. Like other model reduction approaches, this minimal set of nodes is a compromise between available data and model accuracy [15]. Unlike several other reduction approaches, HillTau retains a direct mapping to observable biological entities. Indeed, the HillTau representation of a signaling node may be closer to the conventional intuition based on pathway schematics, than is a full mass-action reaction representation. Like pathway diagrams, each HillTau reaction receives excitatory, inhibitory and modulatory inputs. A further point of similarity is the HillTau models may condense several intermediate steps into a single node on the reaction network. A more subtle point of similarity is that pathway block diagrams typically assume implicit back reactions and decay of activity when stimuli are removed. This too is built into how HillTau reactions work. In contrast to these simple mappings from pathway diagrams to the HillTau form, it is often difficult to map between signaling diagrams and the full mass-action reaction schemes [51,52]. While previous model reduction studies have worked on different pathways than the synaptically biased set explored in our study, a comparison with the survey of methods in [15], suggests that HillTau achieves as good or better model reduction for large models than most other methods.

## System abstraction

Next, *system abstraction* and functional modules help to make sense of complex biological signaling. We propose that HillTau forms a useful tool for arriving at functional modules in complex signaling networks. Such modules have long been considered a conceptual basis for understanding complex signaling [53]. Typically they have been ascertained by manual

inspection and dynamical analysis of components of signaling networks, for example, the nested feedback loops in the cell cycle [54]. A more scalable approach to uncovering such modules is to use graph theory for motif analysis on detailed mass-action models, but this approach loses key aspects of system dynamics [55]. With the HillTau formalism and our procedures for model reduction, we are able to generate highly reduced reaction graphs that nevertheless support rather accurate dynamics. The formalism encourages models that can be readily mapped to biology. ThusHillTau supports a data-driven approach to arrive at functional modules.

While functional modules are good for analysis, we note that biology does not necessarily partition signaling networks into neat modules [52]. Indeed, cross-talk between pathways is common. HillTau supports explicit cross-talk interactions, but does not introduce implicit interactions. In this regard it differs from mass-action reaction systems, in which mechanisms such as back-reactions or enzyme sequestration may introduce subtle effects, such as implicit feedback. For example, one can achieve bistability through multistage phosphorylation/ dephosphorylation cascades [56]. To represent such effects in HillTau one would have to introduce explicit feedback steps between reactions, such as in the bistability example in Fig 2. Similarly, interesting behavior emerging from chemical saturation, such as zero-order ultrasensitivity [57] would require the use of the explicit math expressions supported by HillTau.

Protein-protein interaction networks are a commonly derived form of abstract networks. These can be purely topological, or may also include reaction dynamics [17]. Can these be parsed into HillTau networks? To first order, protein interaction networks lack the distinction that HillTau 'reactions' make between inputs, activators and modulators. Additional information is needed to disambiguate these. Data from sources such as pathway maps, protein domain properties, or Gene Ontology relationships are required to resolve HillTau topologies for a given protein-protein network [10].

Next, can the rates be assigned? In networks that include dynamics (e.g., [17]) this is relatively straightforward to accomplish. In Figs 4 and 5 we illustrate how experimental data, or simulated dynamics of an existing model can be used to parameterize a HillTau model. The same approach could utilize the original timing constraints that went into the Nyman model [17]. Alternatively, a program similar to MASH could explore dynamics of the reference (protein-protein network) model, and use the output to search for parameters of the corresponding HillTau model.

ThusHillTau promotes abstraction through model reduction. The abstracted models expose all interactions explicitly.

## Scaffolds for model fitting

Third, HillTau is a useful intermediate step, or *scaffold*, for model fitting of large mass-action models. Direct model-fitting is difficult in at least two ways: there are typically far fewer experiments than parameters, and it is computationally costly to run a large ODE model many times for carrying out an optimization approach to model fitting. We propose that the HillTau form may provide a useful bridge on both these counts. As we have illustrated in Figs 2, 3 and 5, HillTau models lend themselves to fitting to experiments because they have few parameters and they run quickly. Several advantages accrue from an initial pass to make and fit a HillTau model. 1. In building and optimizing a HillTau model, the optimization dataset will be use-tested, and gaps identified. 2. The essential pathway structure of the model will be defined by the HillTau model, and key interactions identified. The mass-action model must, at minimum, incorporate these interactions. 3. The parameters of the HillTau model set bounds for those of

the detailed reaction sets. For example, the time-course of any individual mass-action reaction step must be faster than the HillTau reaction in which it is embedded.

## Efficient approximations to complex signaling

Finally, HillTau models are useful because they are *efficient*. One of the key use-cases envisioned for HillTau was to model complex cellular signaling, with synaptic signaling as an exemplar. Several of the examples in the current paper are in this domain, specifically the BCM curve (Fig 3A–3E, the coupled BCM curve with bistables (Fig 3F and 3G), and a synaptic protein synthesis pathway (Fig 5). While even these large ODE signaling models run somewhat faster than wall-clock time (Fig 6), there are at least two cases where far greater efficiency is desirable. First, as mentioned above, model parameter optimization requires a large number of evaluations of complete simulations (100s to 1000s in our experience). To perform a single evaluation, these synaptic simulations may have to run for many thousands of seconds of simulation time to compare with typical plasticity experiments [58]. Further, one typically optimizes a pathway to fit numerous experiments, all of which must be simulated for each evaluation. Together, this is computationally expensive. In our HillTau optimizations using MASH, the total simulation time for large numbers of pathway simulations was typically even smaller than the time spent by the minimizer code itself.

A second use case for highly efficient signaling models is in synaptic signaling. A single neuron may have over 10,000 spines, and there may be many such neurons in a network. If each synapse is to implement a complex biochemical pathway the computational costs may be formidable. Network plasticity models [59], and cellular sequence selectivity models [29] are examples of this scale of model. Indeed, Higgins et al. [59] have used an efficient event-driven calculation of synaptic weights with a similar exponential decay calculation as in HillTau. HillTau signaling provides a way to implement biologically detailed synaptic dynamics in every synapse, even in large networks.

In summary, the HillTau form and its supporting toolkits for running and optimizing models provide a compact, efficient way to perform data-driven abstraction of complex signaling models.

## Methods

### HillTau formulation

The HillTau formulation is an event-driven variant of a Hill equation with modifiers. It is specified and evaluated in two stages: the steady-state value, and an exponential time-course of approach to steady-state. As detailed below, reactions in a HillTau model are evaluated in successive layers such that the next estimate for steady-state value of a given layer depends only on boundary conditions and on outputs from preceding layers. In the equations below we expand out the equations for a range of use-cases. In equations 1.x we specify the steady-state values for each use case. In equations 2.x we define the approach to steady state. In equations 3.x we combine Eqs 1 and 2 to summarize the evaluations done in HillTau. In equations 4.x we provide interpretations of the HillTau equations as rate terms which can be evaluated by regular ODE solvers and form the basis for the SBML export of HillTau model systems. However, the definitive form of HillTau is event-driven and the rate-term form should be regarded as a convenient but approximate mapping to conventional mass-action solvers. In equations 5.x we provide a motivation for the form of HillTau as an approximation to complex signaling chemistry.

## HillTau steady-state

The HillTau formulation stipulates that the steady-state level $Y_\infty$ of each signaling step (which may involve multiple chemical steps) is approximated by a Hill function

$$Y_\infty / Y_{input} = \Theta = L^n / (KA^n + L^n) \qquad \text{Eq 1.1}$$

Here $Y_{input}$ is the input concentration to this signaling step, where $Y_{input}$ is either a molecule with a predefined concentration (i.e, a boundary condition) or coming from an upstream reaction. Likewise, the reactant L can either be predefined or come from an upstream reaction. $KA$ is the association constant of L with Y.

We elaborate this slightly to accommodate an optional gain term:

$$Y_\infty = Y_{gain} Y_{input} L^n / (KA^n + L^n) \qquad \text{Eq 1.2}$$

We utilize a slightly modified form to permit the Ligand $L$ to act in an inhibitory manner:

$$Y_\infty = Y_{gain} Y_{input} (1 - L^n / (KA^n + L^n)) \qquad \text{Eq 1.3}$$

In cases where there are modifiers, we include a further term based on the analysis of Hofmeyer and Cornish-Bowden [34]:

$$Y_\infty = \frac{Y_{gain} Y_{input} L^n}{L^n + KA^n (1 + (M/K_{mod})^h) / (1 + A_{mod}(M/K_{mod})^h)} \qquad \text{Eq 1.4}$$

Here M is the concentration of the modifier, $K_{mod}$ is the half-effect value, $A_{mod}$ is the modifier action, and h is the order of the modifier. The modifier acts in an inhibitory manner when $A_{mod} < 1$, and as an activator when $A_{mod} > 1$. When $A_{mod} = 1$, clearly, the modifier has no effect.

Similarly we define the action of the modifier on an inhibitory reaction:

$$Y_\infty = Y_{gain} Y_{input} \left( 1 - \frac{L^n}{L^n + KA^n (1 + (M/K_{mod})^h) / (1 + A_{mod}(M/K_{mod})^h)} \right) \qquad \text{Eq 1.5}$$

We use a different equation to define steady-state behavior of a system where a single substrate molecule $Y_{input}$ is converted into a product $Y$:

$$Y_\infty = Y_{input}^n / KA \qquad \text{Eq 1.6}$$

For the rare cases where a non-chemical formulation is needed to describe the system, we provide an alternative algebraic expression for $Y_\infty$:

$$Y_\infty = f(Y_1, Y_2, \ldots) \qquad \text{Eq 1.7}$$

where f is an arbitrary algebraic function and $Y_1$, $Y_2$... are concentrations of other molecules. The use of this algebraic form is discouraged as it weakens the mapping of the model to the underlying chemistry. This form does not admit of modifiers.

Note that these equations are entirely feed-forward: the concentrations of molecules L, M, and $Y_{input}$ are not affected by their participating in any downstream reactions.

## HillTau time course

Equations 1 define the steady state estimate for molecule Y, given a set of molecule concentrations in the preceding layer. We then assume that the approach of the system to steady-state is

governed by a simple exponential with characteristic time $\tau$ (Fig 1A):

$$\frac{Y(t + \Delta t) - Y(t)}{Y_\infty - Y(t)} = 1 - exp(-\Delta t/\tau) \qquad \text{Eq 2}$$

where Y(t) is the value of Y at time t. For a simple binding reaction, the time course $\tau$ is an experimental observable, and is approximated by $\tau \sim 1/(kf+kb)$ where *kf* and *kb* are the forward and backward rates for the first-order Hill binding reaction (Figs 1 and S1).

As a slight extension to this, we permit an optional separate time course $\tau_2$ when Y is falling:

$$\text{If } Y_\infty > Y(t) \qquad \frac{Y(t + \Delta t) - Y(t)}{Y_\infty - Y(t)} = 1 - exp(-\Delta t/\tau) \qquad \text{Eq 2.1}$$

$$\text{If } Y_\infty < Y(t) \qquad \frac{Y(t + \Delta t) - Y(t)}{Y_\infty - Y(t)} = 1 - exp(-\Delta t/\tau_2) \qquad \text{Eq 2.2}$$

The occurrence of different time-courses for buildup and decay is quite common. It happens in a simple binding reaction (S1A, S1C and S1E Fig). It also occurs when there are different chemical steps, such as enzymes with different rates, mediating competing processes for buildup and decay.

## HillTau composite form

Putting Eq 1 and Eq 2 together, we have the following closed-form expression for the value of Y at time t + Δt:

$$Y(t + \Delta t) = Y(t) + (Y_\infty - (Y(t) - Y_{baseline}))(1 - exp(-\Delta t/\tau)) \qquad \text{Eq 3}$$

The term $Y_{baseline}$ is an optional (positive) baseline level of molecule Y. Δt is the timestep. Note that this is a closed form: Δt can be as large as the end of the simulation.

In the limit of large Δt, we have,

$$Y_{(t=\infty)} = Y_\infty + Y_{baseline} \qquad \text{Eq 3.1}$$

The initial conditions for molecule Y are either specified in the model definition, or as a fallback we estimate the steady-state concentration as in Eq 3.1.

## Rate interpretation of HillTau

Formally, HillTau can be seen as an approximation to the following rate equations:

$$Y'(t) = (Y_\infty - Y(t))/\tau \qquad \text{Eq 4}$$

In cases where we have a separate $\tau_2$, and $Y_\infty < Y(t)$:

$$Y'(t) = (Y_\infty - Y(t))/\tau_2 \qquad \text{Eq 4B}$$

SBML, and most ODE solvers, will not readily handle this switch between $\tau$ and $\tau_2$. For the purposes of the equations below we just use $\tau$.

Expanding out $Y_\infty$, we have five variants of Eq 4:

Basic activation reaction:

$$Y'(t) = \frac{Y_{gain}Y_{input}L^n/(KA^n + L^n) - Y(t)}{\tau} \qquad \text{Eq 4.1}$$

Basic inhibition reaction:

$$Y'(t) = \frac{Y_{gain}Y_{input}(1 - L^n/(KA^n + L^n)) - Y(t)}{\tau} \qquad \text{Eq 4.2}$$

Activation reaction with modifier:

$$Y'(t) = \frac{1}{\tau}\left( \frac{Y_{gain}Y_{input}L^n}{L^n + KA^n(1 + (M/K_{mod})^h)/(1 + A_{mod}(M/K_{mod})^h)} - Y(t) \right) \qquad \text{Eq 4.3}$$

Inhibition reaction with modifier:

$$Y'(t) = \frac{1}{\tau}\left( Y_{gain}Y_{input}\left(1 - \frac{L^n}{L^n + KA^n(1 + (M/K_{mod})^h)/(1 + A_{mod}(M/K_{mod})^h)}\right) - Y(t) \right) \text{Eq 4.4}$$

Conversion reaction:

$$Y'(t) = \frac{(Y_{input}^n/KA) - Y(t)}{\tau} \qquad \text{Eq 4.5}$$

Although the HillTau calculations can be done using equations 4.x with a regular ODE solver, the HillTau definition envisions event-driven calculations. Further, the complete Hill-Tau form uses $\tau_2$ extensively, which is not handled by the above equations. In principle, the optimization for the HillTau model could just use $\tau$ throughout, in which case the reduced HillTau model could be rendered in mass-action form with a reasonable degree of accuracy (S5A Fig).

In Table 1 we summarize the parameters for HillTau. All but the first apply to Reactions.

Simple HillTau models only need species concentrations, and the KA and $\tau$ terms for the reactions. The optional terms greatly facilitate the design goals of compactly specifying diverse signaling reactions.

## Computing time-evolution and steady-states

To build complex reaction systems, we permit cascading of reactions so that any molecule can be a substrate or equation term in any other reaction. To reiterate, this is a purely feed-forward formulation, so substrates are not affected by any of their downstream targets. We obtain a dependency tree so that on each timestep the updates are carried out in an order which ensures that inputs ripple in order through the cascade of reactions. This may lead to inaccurate estimation of transient responses if the updates are carried out at greater intervals (timestep $\Delta t$) than the time-course ($\tau$) of the fastest reaction in the model (S7 Fig). To address this, HillTau assigns an internal timestep $\Delta t$ which is smaller than the smallest $\tau$ in the model. Since one normally performs time-series sampling of reactions at a time finer than the fastest reaction, this restriction usually has little impact on run-time. Further, multi-step systems may include feedback. In such cases the program has to explicitly break the dependency chain. HillTau identifies dependency cycles, picks a reaction based on definition order, and assigns it the next open level in the dependency tree.

A distinct case arises when the HillTau system is used to compute steady state values (e.g., in a dose-response curve). These could ideally be solved by taking an infinitely long time-step.

Given the possible presence of feedback, we instead take a long settling time and subdivide it into 10 equal steps so as to allow feedback reactions to also settle.

## Motivation for the HillTau formalism

The initial motivation for the HillTau form was the observation that many stimulus-response curves in signaling have a saturating, Hill-like concentration dependence on input strength even if there are multiple intermediate steps [4,44]. Further, many stimulus-response time-courses are visually similar to exponential time-courses. This suggested that a combination of these two properties might be a good approximation even to multi-step signaling cascades.

In order to mathematically support this idea, we considered two of the common motifs in signaling: enzyme-activation of molecules such as phospho-proteins, with a balancing deactivation reaction; and binding of activators to a reagent. Below, we show that the HillTau form achieves a reasonable approximation both to the amplitude and time-course of the response.

First, we considered outcomes of an enzymatic cascade with back reactions. We approximate the rate of production using a Michaelis-Menten- form:

$$dP/dt = ES.kcat/(Km + S)$$
Eq 5.1

This is balanced by a first-order back reaction:

$$dP/dt = -Kr.P$$
Eq 5.2

Then the steady-state at dP/dt = 0 is obtained by combining these:

$$P_1 = ES_1\,kcat_1/((Km_1 + S_1)Kr_1)$$
Eq 5.3

This is of the same form as Eq 1, showing that a single enzymatic stage in the cascade can be approximated by HillTau. Here we add a subscript $1$ to indicate that it is the first reaction in the cascade. Now we stipulate that $P_1$ is the catalyst for the next step, substituting for enzyme E2. This stage results in the formation of product $P_2$:

$$P_2 = (ES_1\,kcat_1/((Km_1 + S_1)Kr_1)) \times S_2\,kcat_2/((Km_2 + S_2)Kr_2)$$
Eq 5.4

And so on for multiple steps. Now, suppose that we only have 2 variable inputs to this pathway: the first stage input E and one of the substrates. All other substrates are held fixed. This is a reasonable assumption for HillTau, because any further variable inputs should be treated explicitly either as modulators or as separate reaction steps. Then, all the $(Km_n + S_n)$ terms are constant except the one variable substrate $S_v$. By combining all the constant terms into $K_{cascade}$, we end up with:

$$P_n = ES_v K_{cascade}/(Km_v + S_v)$$
Eq 5.5

This is equivalent to the Hill equation form at the basis of HillTau (See Eq 1.2). We treat inhibition using the same analysis, except resulting in depletion of a substrate (Eq 1.3).

For the time-course, we assume that one of the reactions is rate-limiting. For this step, the rate of formation of product is given by:

$$dP/dt = ES.kcat/(Km + S) - KrP$$
Eq 5.6

This yields an exponential settling curve with a final value $P_\infty$., as shown from Eq 5.3. The time-course is given by:

$$P(t) = P_\infty - (P_\infty - P_0)\,exp(-t/\tau) \qquad \text{Eq 5.7}$$

where

$\tau = 1/kr$ and P0 is the initial value of P.

Note that Eq 5.7 can be rearranged to give Eq 3.

Overall, this approximation yields a consolidated HillTau 'reaction' in which we have one stimulus (E, mapping to the activator L in Eq 1), one reactant ($S_v$ mapping to the reagent $Y_{input}$ in Eq 1), to represent a cascade of enzymatic steps with back reactions.

Next, we consider binding reactions in the pathway. These give the same steady-state Hill Equation form, by definition. From Eq 1.1, setting n = 1, and considering the first reaction generating Y1:

$$Y_1 = L_1 R_1/(L_1 + KA_1) = R_1/(1 + KA_1/L_1) \qquad \text{Eq 5.8}$$

A cascade of similar binding reactions, where Y1 feeds into reaction 2, can also be reduced into the same form:

$$Y_2 = R_2/(1 + KA_2/Y_1) = R_2/(1 + KA_2 \times (1 + KA_1/L_1)/R_1) = R_x/(1 + KA_x/L_1) \quad \text{Eq 5.9}$$

where $R_x = R_1 R_2/(R_1 + KA_2)$ and $KA_x = KA_1.KA_2/(R_1 + KA_2)$

Here Eq 5.9 has the same form as Eq 1.1 and Eq 5.8.

Using a similar approach, any number of cascading binding reactions will end up fitting the same Hill form. Further, from Eq 5.5 we see that the enzyme/back-reaction steps have a similar Hill-like form. Thus they too can be folded into this cascade.

What is the time-course of this cascaded reaction? As before, we assume that the cascade has one rate-limiting step *i*. A standard analysis shows that this too has an exponential time-course.

$$dY_i/dt = Kf_i L_i R_i - Kb_i.Y_i \qquad \text{Eq 5.10}$$

which yields the same exponential settling time-course as Eq 5.7:

$$Y(t) = Y_\infty - (Y_\infty - Y_0)\,exp(-t/\tau) \qquad \text{Eq 5.11}$$

where tau = 1/(L.Kf + Kb) and Y0 is the initial value of Y.

It is important to note that this value of tau has a dependence on a variable, L, and hence the use of a constant value of tau is approximate. This is partially mitigated by utilizing different values of tau for rising and falling phases of the signal Y. During the rising phase, L will have a different (typically larger) mean value than during the falling phase. The use of tau for rising and tau2 for falling phases of the response reflects this.

Thus the steady-state terms for cascading binding and enzymatic reactions can be consolidated into a single HillTau 'reaction' step of the Hill form, and the time-course can be approximated by an exponential when there is one rate-limiting step.

## Model definition and reference implementation

HillTau reaction systems are set up in a simple JSON format (Fig 1C), for which we have provided a schema. We have implemented a small reference driver program in Python (hillTau.py) that loads the model, runs it with optional stimuli, and plots or saves the output. The hillTau.py file also provides a set of library functions for use in larger programs. An equivalent implementation in C++ using PyBind11 for identical Python bindings is also provided. Three

additional utilities are also provided, as described below: for model illustration, model abstraction, and model conversion to SBML. Python scripts for generating the figures in this paper (except Fig 4, which is a schematic only) are also provided in S1 Data and on the GitHub site. Benchmarking was done using the program *fig6.py*, which calls MOOSE and HillTau through their Python interfaces, and calls COPASI through thePyCoTools Python interface [60]. The output values of multiple benchmarking runs were averaged and used for generating the graphs in *fig6_plotting.py*. All HillTau and supporting code is licensed under GPL version 3 or later.

## Model illustration

We developed a utility *htgraph.py*, which generates a reaction diagram for HillTau models specified in the.json format. This diagram gives a complete specification of the model topology, in that one can rebuild the structure of the HillTau model by inspection of the reaction diagram, though of course the parameters are not provided in the diagram. htgraph.py uses the *dot* module of the open-source package *graphviz* [61] to generate the network graphs.

## Model reduction and abstraction: MASH

We provide a utility for performing Model Abstraction from SBML to HillTau: MASH. Briefly, MASH runs the original SBML model with a reference stimulus to explore the key dimensions of its input-output mappings. Typical reference stimuli (built into MASH) include dose-response curves and pulsatile time-series stimuli. MASH then uses standard minimization routines (scipy.optimize library, method "L-BFGS-B", [62]) to tweak the HillTau model parameters to improve its fit to the original model.

MASH is implemented as a Python script *mash.py* which uses an ODE model (SBML) as a reference to which it fits a HillTau model. The user provides an initial HillTau model and specifies a series of stimuli to deliver. As part of this the user also defines which are the input molecules, and which are the readouts, and the ODE and HillTau models. MASH generates a topologically identical HillTau model to the original, with parameters optimized to fit. It also reports initial and final scores, expressed as normalized RMS difference between reference model and HillTau. As per Fig 4, the user may introduce additional intermediate steps in the pathway in order to achieve the target model fit. The user specifies a set of stimuli (typically a combination of dose-response and time-series calculations) that explore the model response space. MASH generates a reference response to these stimuli using an ODE solver (MOOSE). The function evaluation for the minimization is carried out by running the HillTau model with the same stimulus, and comparing the HillTau output point-by-point with the reference. The normalized RMS difference over all points is returned as the score. A score of below 0.05 means that on average the original response differs from the HillTau response by less than 5%. MASH uses the scipy.optimize library to tweak the HillTau model parameters to improve the fit, as measured by this RMS score. MASH documentation and examples are provided on the HillTau website. MASH was used to fit the HillTau models for Figs 2, 3 and 5, and the scores are reported. S2, S3 and S4 Figs, and S1 Data specify how these fits were done.

## Model conversion to SBML

The utility *ht2sbml.py* performs a conversion of HillTau models defined in the reference JSON format, into equivalent ODE models defined in SBML. It uses *simplesbml* (https://github.com/sys-bio/simplesbml for generating the SBML. This uses the forms defined in equations 4.x in Methods. The conversion is approximate on two counts, first, HillTau is an event-driven, not continuous method, and second, HillTau may use different time-courses for rising and falling

phases as a reaction proceeds, whereas the ODE form uses only a single time-course. If the HillTau model is generated (for example, after model reduction) such that each reaction only utilizes tau and not tau2, then the approximation is very good. In S5 Fig, we performed *ht2sbml.py* conversion of 3 HillTau models to SBML and then compared the HillTau output with COPASI calculation of the converted model, under 5 conditions. We obtained an excellent fit (<1% normalized RMS) for an oscillatory model that did not use tau2. The feedback-inhibition model used in Fig 2, which does use tau2, gave a mediocre fit of 29%. The full protein synthesis model was tested under 3 conditions: protein response to BDNF, S6K response to BDNF, and CaMKIII response to Ca. These gave fits of 30%, 26% and 1.7% respectively, though the qualitative response was similar in all cases. Thus the *ht2sbml.py* conversion works for all HillTau models, but the conversion may be approximate for models which have very different tau and tau2 parameters in their reactions.

## Supporting information

**S1 Fig. HillTau fits to simple mass-action reactions.** Fits are indicated on top of each figure panel. Each of these is a single HillTau 'reaction' where 'input' is activator in all but Panel E, where 'input' is an inhibitor. In all cases the rising phase fits exactly, but in panels A, C and E the falling phase has a different time-course.
(TIF)

**S2 Fig. Fit of single-reaction HillTau Model to protein synthesis pathway.** Model is as in Fig 5B. Panels A-F correspond to panels D-I in Fig 5. In all cases protein production rate is readout. Blue plots are reference, orange areHillTau. A: BDNF@3.7 nM + $Ca^{2+}$@0.2 μM, 900 seconds. B: BDNF@3.7nM, $Ca^{2+}$@1μM. C: 3 pulses of BDNF@3.7 nM for 5s, coincident with $Ca^{2+}$@10μM for 1s, pulses separated by 300 s. D: Same as C, but $Ca^{2+}$ held at baseline of 0.08 μM. E: Dose-response of protein vs. $Ca^{2+}$, holding BDNF fixed at 3.7 nM. F: Dose-response of protein vs BDNF, holding $Ca^{2+}$ fixed at 0.08 μM. G: MASH optimization waveform used to fit the HillTau model for protein synthesis.
(TIF)

**S3 Fig. Fitting S6K to the protein synthesis pathway model.** HillTau reactions as in Fig 5C. Panels A-F correspond to panels D-I in Fig 5. In all cases activated S6K concentration is readout. Blue plots are reference, orange areHillTau. A: BDNF@3.7 nM + $Ca^{2+}$@0.2 μM, 900 seconds. B: BDNF@3.7nM, $Ca^{2+}$@1μM. C: 3 pulses of BDNF@3.7 nM for 5s, coincident with $Ca^{2+}$@10μM for 1s, pulses separated by 300 s. D: Same as C, but $Ca^{2+}$ held at baseline of 0.08 μM. E: Dose-response of protein vs. $Ca^{2+}$, holding BDNF fixed at 3.7 nM. F: Dose-response of protein vs BDNF, holding $Ca^{2+}$ fixed at 0.08 μM. G: MASH optimization waveform used to fit the HillTau model for S6K activation.
(TIF)

**S4 Fig. Fitting CaMKIII to the protein synthesis pathway model.** HillTaureactions as in Fig 5C. Panels A-F correspond to panels D-I in Fig 5. In all cases activated CaMKIII concentration is readout. Blue plots are reference, orange areHillTau. A: BDNF@3.7 nM + $Ca^{2+}$@0.2 μM, 900 seconds. B: BDNF@3.7nM, $Ca^{2+}$@1μM. C: 3 pulses of BDNF@3.7 nM for 5s, coincident with $Ca^{2+}$@10μM for 1s, pulses separated by 300 s. D: Same as C, but $Ca^{2+}$ held at baseline of 0.08 μM. E: Dose-response of protein vs. $Ca^{2+}$, holding BDNF fixed at 3.7 nM. F: Dose-response of protein vs BDNF, holding $Ca^{2+}$ fixed at 0.08 μM. G: MASH optimization waveform used to fit the HillTau model for CaMKIII activation.
(TIF)

**S5 Fig. Conversion of HillTau models to SBML, and comparison of the resultant responses simulated in HillTau and COPASI respectively.** A: Oscillator model. This uses only 'tau' in its formulation, and fits to within 1%. B. Feedback inhibition model from Fig 2. A 1 uM stimulus is delivered at t = 20, and it lasts till t = 60. This has a mediocre fit of 29%. C-E: Protein synthesis model. C. Comparing protein synthesis response to a BDNF stimulus of 5 nM from t = 2000s to t = 3000s. Fit = 30% is mediocre. D. S6K activation in response to a BDNF stimulus of 5 nM from t = 2000s to t = 3000s. Fit = 26% is mediocre. E. CaMKIII activation in response to a calcium stimulus of 5 uM from t = 2000 to t = 3000s. This fits well, 1.7%. F. HillTau reaction scheme for oscillator model.
(TIF)

**S6 Fig. HillTau model schematic for largest model in Fig 6, with 35 HillTau parameters and 11 reactions.**
(TIF)

**S7 Fig. Dependence of HillTau simulation output on timestep.** In all panels the dashed lines represent the time-series, and the dots represent the sample points for estimating error using the smallest timestep as reference. Accuracy is reported as normalized root-mean square difference from smallest timestep. A: Feedback inhibition. Step stimulus of 1 uM is given at t = 10s, which lasts till t = 50s. 1% accuracy is achieved for dt = 1s. B: feedforward inhibition. Stimulus same as A. 1.5% accuracy at dt = 1s. C: BCM curve. Stimulus of 1 uM is given at t = 10s and stays till the end of the simulation. 1% accuracy at dt = 1s. D: Kholodenko oscillator. Here the system is free-running. 1.2% accuracy at dt = 6s.
(TIF)

**S1 Data. Supplementary code directory.** The zipfile S1 Data with supplementary code expands out into a directory which has Python scripts and model definition files for generating all the simulated components of the figures in the paper, including supplementary figures. Detailed instructions for running the scripts are provided in the README.txt file in the same directory. The scripts should run with Python 3.x, but many figures require additional software installation for the ODE simulators MOOSE and COPASI, as well as some other packages. Details are provided in the README.txt.
(ZIP)

## Acknowledgments

Nisha Viswan provided useful feedback from use-testing HillTau. G.V. HarshaRani did much of the coding for htgraph.py and configured the HillTau repository for installation using pip.

## Author Contributions

**Conceptualization:** Upinder S. Bhalla.

**Data curation:** Upinder S. Bhalla.

**Formal analysis:** Upinder S. Bhalla.

**Funding acquisition:** Upinder S. Bhalla.

**Investigation:** Upinder S. Bhalla.

**Methodology:** Upinder S. Bhalla.

**Project administration:** Upinder S. Bhalla.

**Resources:** Upinder S. Bhalla.

**Software:** Upinder S. Bhalla.

**Supervision:** Upinder S. Bhalla.

**Validation:** Upinder S. Bhalla.

**Visualization:** Upinder S. Bhalla.

**Writing – original draft:** Upinder S. Bhalla.

**Writing – review & editing:** Upinder S. Bhalla.

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
