## [Decision Letter · Decision Letter 0]

18 Aug 2021

Dear Prof. Bhalla,

Thank you very much for submitting your manuscript "HillTau: A fast, compact abstraction for model reduction in biochemical signaling networks" for consideration at PLOS Computational Biology. As with all papers reviewed by the journal, your manuscript was reviewed by members of the editorial board and by several independent reviewers. The reviewers appreciated the attention to an important topic. Based on the reviews, we are likely to accept this manuscript for publication, providing that you modify the manuscript according to the review recommendations.

Sincerely,

Joanna Jędrzejewska-Szmek, Ph.D.

Guest Editor

PLOS Computational Biology

Kim Blackwell

Deputy Editor

PLOS Computational Biology

[LINK]

Reviewer's Responses to Questions

**Comments to the Authors:**

Reviewer #1: Report on the paper "HillTau: A fast, compact abstraction for model reduction in biochemical signaling"

This paper introduces a new methodology and tool for simulating signaling networks. I find the approach an interesting

alternative to mechanistic ODE models. As the author clearly points out, HillTau has less parameters than mechanistic models and

the execution time is several orders of magnitude faster. These crucial advantages are

appropriately exploited by the machine learning features of the present tool.

However, I find the presentation of the tool and methodology rather expeditious. The mathematical foundations are far from

clear and may lead to confusion. The relation with other abstraction methods is also not clearly presented.

Given the added value of the tool and methodology I recommend publication, provided that the following points are clarified:

1) Introduction.

a) The most current approach in machine learning of dynamical signaling networks is based on neural network type models employing

sigmoidal functions (see for instance Nyman et al Plos Comp Biol 2020, PMID: 32667922). A parallel/comparison to these approaches should be drawn.

b) The fast reaction detection invoqued at line 72 is a particular case of more general quasi-equilibrium and quasi-steady state

based model reduction methods reviewed in Radulescu et al Frontiers in Genetics 2012, PMID: 22833754. These reduction methods use graph rewriting and

retain indeed the chemical reaction network formalism as mentioned at line 76.

2) Hilltau schemes.

The figures and examples are useful for understanding what a Hilltau model is, but are not sufficient. I was not able to find in the paper

an algorithmic description about how to generate a Hilltau scheme given a full mechanistic model. Are these abstractions obtained

manually? It would be also interesting to discuss the generation of Hilltau models starting with protein-protein interaction

networks that are available in larger quantity and scale than the mechanistic models.

3) Testing accuracy.

The author has presented the results of extensive benchmarking of his tool. However, I would have liked to see extensive data

not only on the efficiency (Figure 6) but also on the accuracy of HillTau.

4) Domain of validity.

Several properties of mechanistic models are necessarily lost by this approach. For instance, even if multistationarity with

feedback can be reproduced, multistationarity without feedback (resulting from sequestration effects, see for instance Markevich et al J Cell Biol 2004

PMID: 14744999) is lost. There is a bit of confusion here concerning conservation of the steady states. The method conserves steady states

to some extent, but not stoechiometry classes; this subtle difference leads to the lost of multistationarity without feed-back.

It would be useful that the author warns about other properties that may be lost by using

HillTau abstraction.

5) Methods.

a) definitions should be provided for different concepts, for instance input concentration to a signaling step, level of a signaling step, ligand, modifier, etc.

b) It is not clear why Eq 1.7 does not admit modifiers (line 599)

c) rate interpretation of HillTau (lines 637 - 670). I could understand this heuristics, but here one should notice that the ligands and modifiers are considered

constant in the rate interpretation. This assumption may work for small time steps, but not for large time steps. The author seem to care about the size

of the time step only in the presence

of feedback, whereas large time steps may lead to inacurrate dynamics also without feedback. The main issue here is that the dynamics of one variable in a

network is not mono-exponential, but multiexponential. Only when the timescales of the variables are well separated, single variables have mono-exponential

dynamics. This remark can be related to the previous point 4).

d) motivation for the HillTau formalism (lines 688-747). This part of the methods should be entirely revisited as it is neither clear nor rigorous.

Reviewer #2: The author proposes a new coarse grained modeling approach using equations the author calls HillTau which incorporates Hill like equations coupled to an exponential term to model time delays. There could merit in this approach as there is a significant problem in being able coarse grain signaling models and a solution to this problem is needed. I would like to see one thing added to the paper near the beginning which is a step-by-step example of how someone could take, say the MAPK pathway with feedback and convert it into a HillTau model. I note that the author has conveniently provided two python utilities to convert to and from HillTau models and SBML. I wonder if the simulation tooling the author provides is even needed because once in SBML it can be modeling by many other applications, but that is more of an observation than anything else. Finally, the authors states:

“It is also possible to use linear algebraic root-finding to find the steady-state value in one step, but this is prone to numerical challenges even in modern simulators like COPASI”

I don’t believe this. The author cites Clarke 1981 to support this claim but since that publication a lot of work has been done on the problem and so long as the initial starting point is not too far away from the solution, using newton solvers to find the steady state invariably works. NLEQ and Kinsolv are two very robust newton solvers. I agree that problems will arise if there are conserved moieties in the network which are common in signaling pathways. However, this is easily taken care of by doing a row reduction of the stoichiometry matrix first. One can even do a short presimulation (which many of the mainstream simulators do) to give the solvers a good starting point. I am happy to be corrected if the author can find more recent publications that suggest there are problems.

Reviewer #3: This manuscript describes a framework, dubbed HillTau for representing reaction-diffusion systems that is at once both familiar and novel. The framework is familiar in that it is based on Hill equations for steady states and biologically meaningful time constants. The novelty lies in the event-based simulation this approach enables, as the time of the next transition can be calculated using simple arithmetic expressions. This approach vastly reduces the number of parameters relative to mass-action kinetics by avoiding the need to simulate every interaction and instead focus on the relationships between key concentrations. The potential benefit from reducing the number of parameters and avoiding the need for integration through continuous time is vast as this will allow models to incorporate reaction-diffusion based kinetics at scale with minimal computational overhead. The discussion highlights four potential areas of application: model reduction, system abstraction, scaffolds for model fitting, and efficient approximations to complex signaling.

I think this is mostly a very promising paper with an associated well-documented GitHub repository, however I do have two larger concerns: (1) I'm not sure how we're supposed to interpret quality of fit, (2) the order of the paper was confusing with the HillTau model not being explained or motivated until the methods section.

With respect to quality of fit, the main metric used here is RMS error, which gives promising values, but visually the orange (HillTau) and blue (original) look fairly different to me in Figures 3C, 3D, 5D-G, and 5J. In 5G, there is a systematic bias, with the HillTau solution always below the mass-action version. Curiously, there is even visible difference in 2C, which has fairly smooth kinetics. To the extent that the output of these kinetics may feed into some non-linear system, these variations could be significant. The fits may very well be good enough for modeling, but this is not currently self-evident.

As far as the order goes, the results section opens with a brief description of HillTau, but until getting to the methods, there was no concrete example or equation, making it hard to understand what exactly the approach is that's being described and how it allows event-based simulation. The multiple time constants mentioned in line 326 had not been previously mentioned, leading to additional confusion about the nature of the method. All-in-all, the paper does a good job describing the method, but it could benefit from some of that description and motivation happening earlier. Perhaps the HillTau formalization could be thought of as a result, with the mehtods being mostly e.g. how to fit?

Minor:

It is not clear what the numbers in e.g. Figure 2F and 2G (and similarly elsewhere) mean.

Given that a key strength is that the time constants are supposed to be biologically meaningful, it's too bad that the models are being fit using scipy optimization. Is there a principled way to derive the time constants from the mass-action rates?

In at least some of the examples e.g. Examples/PaperFigures/bench_native.py, elapsed time is measured using deltas of time.time(), which gives timings to the nearest 1/64 sec. time.perf_counter() gives higher resolution times that are not susceptible to e.g. changes in system clock. See https://www.webucator.com/article/python-clocks-explained/ for a discussion.

Line 345: "setused" -> "set used" (missing space)

**Have the authors made all data and (if applicable) computational code underlying the findings in their manuscript fully available?**

Reviewer #1: Yes

Reviewer #2: Yes

Reviewer #3: Yes

PLOS authors have the option to publish the peer review history of their article (what does this mean?). If published, this will include your full peer review and any attached files.

Reviewer #1: No

Reviewer #2: No

Reviewer #3: No

Figure Files:

Data Requirements:

Reproducibility:

References:

---

## [Decision Letter · Decision Letter 1]

12 Oct 2021

Dear Prof. Bhalla,

Thank you very much for submitting your manuscript "HillTau: A fast, compact abstraction for model reduction in biochemical signaling networks" for consideration at PLOS Computational Biology. As with all papers reviewed by the journal, your manuscript was reviewed by members of the editorial board and by several independent reviewers. The reviewers appreciated the attention to an important topic. Based on the reviews, we are likely to accept this manuscript for publication, providing that you modify the manuscript according to the review recommendations.

Sincerely,

Joanna Jędrzejewska-Szmek, Ph.D.

Associate Editor

PLOS Computational Biology

Kim Blackwell

Deputy Editor

PLOS Computational Biology

[LINK]

Reviewer's Responses to Questions

**Comments to the Authors:**

Reviewer #1: The revised version of the manuscript addresses all my recommendations and can be accepted.

Reviewer #2: No further comments on the paper.

Reviewer #3: I thank the author for his responses, which largely address my concerns. This manuscript has been strengthened by his edits, especially the addition of a summary of the nature of the HillTau model at the beginning of the results section. In brief, HillTau can be viewed as a phenomenological model that lumps multiple reactions together. Key features are its event-driven nature which allows simulations to make relatively large advances as compared to differential-equation based models, and asymmetric exponential change. Feedback cycles are not directly compatible with the model, but can be approximated by breaking the cycle and using small advances.

Although, honestly, I'm still a little confused because the approach uses an internal time-step so it can't skip from event to event.

The existing implementation on GitHub is pure Python, but the readme promises a faster C++ version that preserves the existing Python API. This will be invaluable.

I still wonder how I would decide if a simplified HillTau model was good enough for my purposes, but the authors points in lines 397-406 (all line numbers here and below are in the version with edits visible) do a good job of addressing the tradeoffs inherent in building "compact models."

Minor:

lines 183-184: A factor of 10 usually gives good convergence. <-- would benefit from knowing what evidence this was based on

MASH is introduced in lines 195-197 , used in 209-210, then seemingly reintroduced in 358-361... would be better if this was consolidated

line 357: readers may not be familiar with FindSim (Viswan et al. 2018); the paper would benefit from a brief description... it's not clear why that approach is best here vs other approaches

lines 457-458: not 100% clear what you're considering state variables here; usually these would be things that have associated rates of change, and then you very much would want to know the initial values. I'm thinking of things like gating variables in Hodgkin-Huxley. There's no initial concentration but there's definitely an initial value that matters.

Line 469: mysterious space in the middle of C++.

Line 914: I think "HillTaul" is a typo, otherwise I missed something major.

**Have the authors made all data and (if applicable) computational code underlying the findings in their manuscript fully available?**

Reviewer #1: None

Reviewer #2: Yes

Reviewer #3: Yes

PLOS authors have the option to publish the peer review history of their article (what does this mean?). If published, this will include your full peer review and any attached files.

Reviewer #1: No

Reviewer #2: No

Reviewer #3: No

Figure Files:

Data Requirements:

Reproducibility:

References:

---

## [Decision Letter · Decision Letter 2]

8 Nov 2021

Dear Prof. Bhalla,

We are pleased to inform you that your manuscript 'HillTau: A fast, compact abstraction for model reduction in biochemical signaling networks' has been provisionally accepted for publication in PLOS Computational Biology.

Best regards,

Joanna Jędrzejewska-Szmek, Ph.D.

Associate Editor

PLOS Computational Biology

Kim Blackwell

Deputy Editor

PLOS Computational Biology

Reviewer's Responses to Questions

**Comments to the Authors:**

Reviewer #3: The author has addressed my concerns, and I have no further comments.

**Have the authors made all data and (if applicable) computational code underlying the findings in their manuscript fully available?**

Reviewer #3: Yes

PLOS authors have the option to publish the peer review history of their article (what does this mean?). If published, this will include your full peer review and any attached files.

Reviewer #3: No

---

## [Editor Report · Acceptance letter]

24 Nov 2021

PCOMPBIOL-D-21-01193R2 

HillTau: A fast, compact abstraction for model reduction in biochemical signaling networks

Dear Dr Bhalla,

I am pleased to inform you that your manuscript has been formally accepted for publication in PLOS Computational Biology. Your manuscript is now with our production department and you will be notified of the publication date in due course.

With kind regards,

Katalin Szabo
